

# A Guide of Indicators Creation for Critical Infrastructures Resilience. Based on a Multi-criteria Framework Focusing on Optimisation Actions for Road Transport System

Zhuyu Yang[1,2], Bruno Barroca[1], Ahmed Mebarki[3,4], Katia Laffréchine[1], Hélène Dolidon[5], Lionel Lilas[6]

[1]Lab'urba, Université Gustave Eiffel, Champs-sur-Marne, 77420, France
[2]LATTS, UMR CNRS 8134 Universit´e Gustave Eiffel/Ecole des Ponts ParisTech, Marne la Vallee, France
[3] University Gustave Eiffel, UPEC, CNRS, Laboratory Multi Scale and Simulation (MSME/UMR 8208), 77454 Marne-la-Vallée, France
[4]Nanjing Tech University (China), Permanent Guest Professor within "High-Level Foreign Talents Program" grant
[5]CEREMA, Nantes, France
[6]DIR Ouest (DIRO), Nantes, France

*Correspondence to*: Zhuyu Yang (zhuyu.yang@univ-eiffel.fr)

**Abstract.** Criteria and indicators are frequently used for assessing the resilience of Critical Infrastructures (CIs). However,
the application of the concept of CIs resilience in practical disaster management is challenged by the lack of operational
tools. An operational tool should enable the establishment of an organized system of indicators and optimising operational
practice. Therefore, to address the operationalisation of resilience assessment, the main objective of this study is to develop a
step-by-step guide for the creation of specific indicators aimed at different practical situations. This guide can assist CIs
managers in their decision-making as it is structured based on a multi-criteria framework that considers the various interests
of stakeholders. This guide includes the methods for Criteria and indicators setting, reference definition, and data collection.
Furthermore, this study presents an example of the application of the guide. This example is based on a given scenario
focusing on the Nantes Ring Road (NRR) network: when it is flooded and closed, the road network manager suggests
alternative roads to citizens. The created indicators, based on this scenario and involved 62 676 data, relate to potential
damages and costs-benefit and involve technical, social, and environmental dimensions.

**1 Introduction**

The research for Critical Infrastructures (CIs) goes across disciplines, sectors, and scales due to the interdependences and
connections between them and other components in the human environment, which consists of physical, social, and
economic conditions and factors affecting human activities and living. Modern infrastructures are the technological systems
that imply a certain heterogeneity of sub-entities-hardware elements also non-physical components, as infrastructures are
formed when engineered systems and socio-ecological context are integrated (Mottahedi et al., 2021). However, CIs are



vulnerable to natural and technological disasters worldwide. Resilience, presented as an inherent attribute of a system addressing external hazards, has developed rapidly in the last decades.

Meanwhile, resilience assessments have become key aspects of CIs management. An efficient resilience assessment could integrate a set of key concepts and provide alternative ways of thinking about and practicing resource management (Resilience Alliance). Indicator-based resilience assessment could be simply summarised as a process consisting of three factors and two phases, as shown in Fig. 1 (Yang et al., 2023b). The principal of indicator-based assessment is transforming from data, to indicators and from indicators knowledge or goal. Available methods for both phases of resilience and indicators assessment are diverse and multidisciplinary, and could be quantitative, qualitative and semi-quantitative

(Mebarki, 2017).

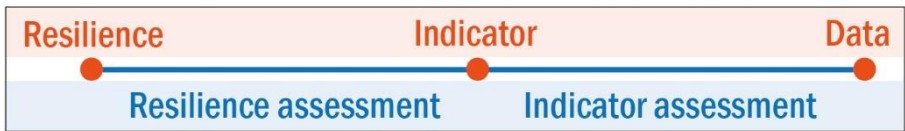

Fig. 1. Indicator-based Resilience Assessment, source: Yang et al. (2023b).

Current studies for CIs resilience are highly related to engineering (Cutter et al., 2010)or socio-technical science (Smith and Stirling, 2010). The studies of the CIs resilience aims to develop more effective and sustainable infrastructure management

strategies for CIs through the concept of "resilience". In other words, one of the desired developments in resilience research is to put resilience-based theories, tools and models into practice and make them useful and operational in disaster management. However, the resilience of CIs is facing challenges in terms of operationalisation in practice (Hosseini; 2016; Meerow et al., 2016; Hernantes et al., 2019; Heinzlef et al., 2022; Esmalian et al., 2022; de Magalhães et al., 2022; Barroca et al, 2023). Operationalising the concept of "resilience" will be a major milestone that contributes to the hazards

management for CIs, even for cities, and the interactions required to build and sustain it.

A pre-analysis based on some current studies shows: even though existing CIs resilience assessments by indicators are diverse and multidisciplinary, it lacks general methods that help CIs stakeholders to create specific indicators regarding concrete situations. As augured by Shavelson et al. (1991) "no indicator system could accommodate all of the potential

indicators identified by a comprehensive process and still remain manageable". A desirable hazard-related indicators tool enables users to create their own personal list of indicators, taking into account their specific situation, without providing directly pre-defined indicators (Barroca et al., 2006). Therefore, the first objective of the present study is to provide a guide for consulting potential users to identify indicators adapted to different concrete situations.



In addition, indicator creation should rely on determined criteria that serve as characters or signs making a judgment of appreciation. From an operational perspective, multi-criteria analysis allows CIs managers to keep holistic thinking that balances the various advantages and disadvantages (Yang et al., 2023a). However, multi-criteria assessments are inadequate in ongoing CIs resilience studies. In particular, much of the research focuses on the abstract capabilities associated with resilience but overlooks the fact that it is vital for every CIs manager to discuss effective actions that can be implemented

without excessive cost or negative impact. The lack of discussion about the strengths and weaknesses of implementation actions creates equally the difficulties in the operationalisation and practice of CIs resilience assessment. It requires, therefore, a multi-criteria analysis involving optimisation actions before indicators creation.

  In order to make resilience assessment integrated into operational disaster management, the present study aims at designing a

guide for creating specific indicators based on a criteria framework addressing optimisation actions. The first step is to discuss the indispensable keys that enable indicators creation and to ascertain their definitions and conceptualizations (section 2). Section 3 designs a step-by-step guide that enables users to create specific indicators to suit their particular situation. Section 4 will illustrate how to create indicators through an example, and section 5 shows a comprehensive assessment process (including resilience and indicator assessment phases in Fig.1) based on created indicators. The example

relied on Nantes Ring Road (NRR) system with the participation of an infrastructure management organisation-Direction interdépartementale des routes Ouest (DIRO) in charge of the road networks of Nantes City in France. This application example involves 62 676 data for traffic flow from DIRO and more than 15 000 data of road infrastructures from BDTOPO of National Geographic Institute (IGN).

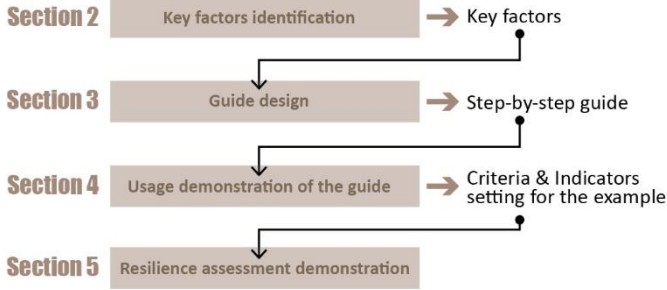

**Fig. 2. Methods and structure of the present study.**

## 2 Keys for Indicators Creation

### 2.1 Criteria Setting

  Suitable indicators are created based on selected assessment criteria, which could be determined through studied goal phenomena, aspects and observed factors (Maggino, 2017). Criteria are characters or signs, which make it possible to

distinguish a thing, or a concept, to make a judgment of appreciation. Each indicator is associated with a criterion, whereas a



criterion is associated with a number of indicators. Criteria could be considered as the points to which the information provided by indicators can be integrated and where an interpretable assessment crystallises. In order to make judgements, different levels of each criterion are generally determined to show what is achieved, how much is accomplished and to what extent. In the field of CIs, stakeholders or managers defined frequently the function of target infrastructure as a criterion. For instance, more than one indicators could assess the function of a road network: number of passing vehicles, vehicle speed, or types of vehicles accepted.

Assessments consisting of criteria and indicators provide a commonly agreed framework for articulating and defining expectations or targets. There is hierarchical structure (Fig. 3), firstly developed for forest sustainability assessment (Prabhu et al.,1996; Lammerts Van Bueren et al., 1997; Mendoza and Prabhu, 2000), today is also used in other disciplines (Montaño et al., 2006; Van Cauwenbergh et al., 2007; Koschke et al., 2012; Feiz and Ammenberg, 2017). This hierarchical structure is a common framework, in which a higher-level "goal" is divided into aspects or themes, which are in turn divided into criteria each with a number of indicators (Maggino, 2017). The assessment process is from "indicators" to "goals", but the criteria and indicators are set in the opposite direction, i.e. based on the phenomena and definition of the studied goal and aspects (Eurostat, 2014; Maggino, 2017). In practical management, the criteria may vary between different contexts. To practice the objective guide, the chosen criteria contributing to indicators creation should enable making criteria specific for adapting to particular situations.

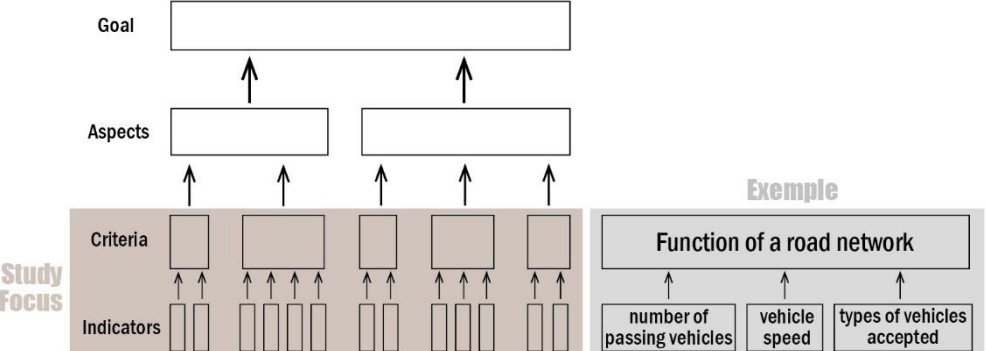

**Fig. 3. Hierarchical structure in multi-criteria approaches for C&I-based assessment, adjusted from Yang et al. (2023).**

The integration of optimisation action into assessment criteria is one of the keys for resilience operationalisation (Yang et al., 2023a). The objective of resilience characterisation and assessment is for helping CIs managers find measures or actions more sustainable and efficient to practically deal with hazards. A resilient CI should have different aspects of capabilities and involve actions to improve its capabilities (Barroca and Serre, 2013). The implementing action refers to all possible operations that could be taken for optimising the resilience of CIs, like programs, strategies, projects, measures, or practices for both temporary (short-term) and permanent preventive (long-term) management. Meanwhile, the optimisation actions aiming at one CI potentially bring unexpected negative effects (like side effects or over-budget expenses) itself or externally



connected systems. Thus, CIs managers can benefit from the results of indicators assessment to recognise effective actions, which produce few costs or negative impacts. Thinking about spatial and temporal interactions of implementation actions, across urban systems, helps enhance beneficial strategies and suppress dangerous ones.

## 2.2 Indicators Setting and Indicators References

According to Cambridge Dictionary, an indicator is a sign or signal that shows something exists or is true, or that makes something clear. Indicators are objective information. A single indicator can rarely provide useful information (Shavelson, 1989). Generally, it is a collection of indicators that help to present the most important and relevant features of a given issue or topic (Eurostat, 2014). Indicator-based assessment consists of setting the expected evolution for the indicator by reference. For each created indicator, appropriate reference values, requested by users and helping to interpret the method's results (Acosta-Alba et al., 2011), should be available for a comparative evaluation (Franchini and Bergamaschi, 1994). "The determination of reference values, norms, or veto thresholds constitutes a key stage in the procedure for developing an indicator. Reference values help to interpret the indicator value and may guide the evolution of a system towards an acceptable level defined in the objectives of the study" (Acosta-Alba et al., 2011). The reference of an indicator can be used as a ruler for measuring a criterion by this indicator, with a scale marked on it. We take the example indicator just mentioned, "number of passing vehicles" and "types of vehicles accepted". For the former, for instance, high function refers to more than 10 000 passing vehicles by day, while low function refers to less than 500 passing vehicles. For the second indicator, a high function of a road network means that all types of vehicles could enter the road network, whereas low function means the network are available for motorbikes only. It can be seen that the definition of the indicator reference also includes the choice of object, unit, and types of attributes (quantitative and qualitative).

Defining a desirable state of resilient CIs is not simple. References for indicators differ according to local conditions. For example, in France, a document named "Atlas of the flood zones of the Loire Valley" (Les services de l'Etat en Loire-Atlantique) aims to provide local authorities and the public with information on historical flood risk in the form of text and maps. A flood hazard map defined on the basis of the Highest Known Waters (PHEC) determines zoning according to the intensity of the phenomenon that can be observed. This map distinguishes three classes of hazards according to the height of submersion defined as follows:

- low hazard: submersion height less than 1.00 m.
- medium hazard: lower submersion height between 1.00m and 2.00 m
- strong and very strong hazard: submersion height greater than 2.00 m.

In this example, hazard intensity is a criterion, while the height of submersion is an indicator. The hierarchical description of these three classes of hazards is a reference to the indicator. This reference standard is highly relevant to the context of the situation of relevant areas. It could be suitable only for France, not for other countries or cities. Alternatively, on the other




hand, the reference does not apply once the invested target has changed. For a road network, a submersion more than 30cm

signifies a catastrophic damage (Liu et al., 2022) (Fig. 4).

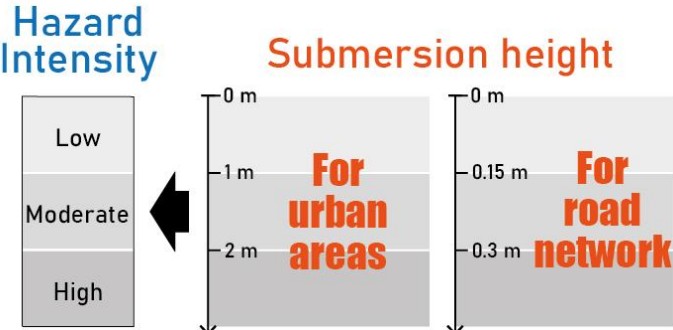

**Fig. 4. Same criterion and indicators could have different references in regarding studied target.**

## 150 **2.3 Data Collection**

The definition of relevant references is highly related to the availability of local data. An indicator, with a reference that cannot be indicated by existing data, loses significant practicality because the manager will not create immediately a new database or a new type of information for one indicator. Data is a discrete fact, a raw element, or the result of an observation, an acquisition, or a measurement, carried out by a natural or artificial instrument. Traditional data types: numerical, text,

graph, Web, and image (Han et al., 2011). The current tendency, big data, which could be categorised by collection technic: Satellite imagery, Aerial imagery and videos, Wireless sensor web network, Light Detection and Ranging (LiDAR), Simulation data, Spatial data, Crowdsourcing, Social Media, Mobile GPS and call record (Sarker et al., 2020).

Meanwhile, data collection is one of the most important parts of constructing indicators (Balaei et al., 2018). Indicator

measure needs reliable data (Vogel, 1997; Cutter, 2016; CORDIS-Smart Resilience Indicators for Smart Critical Infrastructures, 2018). Even though data are objective and do not have to function to evaluate or assess an object, the difficulties of data collection, like lack of unity on definitions, and lack or deficiency of data (Balaei et al., 2018) impact indicators assessment. Thus, it requires also ensuring that there is sufficient data associated with the selected references for indicators. Prabhu et al. (1999) believe even that the difficulty and cost-effectiveness of data should be taken into account in

the evaluation of the indicator's confidence. Local institutions engaged in data collection realise their tasks always based on existing references. This means that when we rely on existing resources to define a reference, we can also find data that we can use. For example, many available data correspond to the reference of the height of submersion mentioned above (i.e. 1.00m and 2.00m). The levels of submersion are oriented by the Ministry of Ecological Transition of France, for providing a concrete, visual, and precise element on the threat of major flooding on a large number of rivers in France.




## 3 Guide for Indicators Creation

The process of indicators creation follows criteria setting and involves reference definition. The former relies on managers' knowledge of the target infrastructure, while the latter necessitates investigation of references and data appropriate to the target infrastructure. It can be argued that the creation of indicators depends on the local human (managers) and material resources (documents, data, terms, etc.) of the studied infrastructures. Indicator-based assessment could be understood as a process leading local resources to an operational tool for disasters management. In this assessment procedure, the creation of indicators, containing reference definitions, can be seen as a bridge grounded in "criteria definitions" and "available data analysis" (Fig. 5). The objective of the present study, designing a guide for creating indicators, is not easy if it requires considering criteria and available data at the same time. A practical guide should be step-by-step and hierarchical. Therefore, section 3 suggests creating firstly possible indicators and their reference based on the settled criteria and then selecting applicable indicators according to available data.

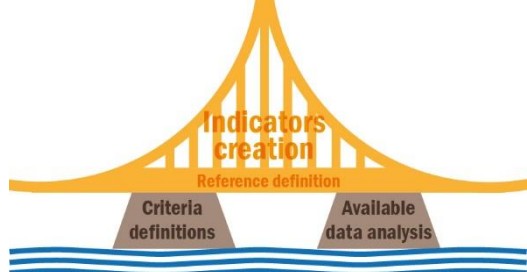

**Fig. 5. Key factors for indicators creation.**

In summary, the guide needs to include three steps:
1.      Specific criteria definition
2.      Possible indicator creation and reference definition
3.      Indicators selection through the availability of data

### 3.1 Specific Criteria Definition

Combining the grounds described in section 2.1, the Multi-Criteria Framework (MCF) developed by Yang et al. (2023) aiming at CIs resilience assessment is deployed for criteria creation for two reasons. Firstly, through an analysis of the definition, phenomena and key aspects of the concept of CIs resilience, this MCF defined four general criteria, "damage of internal components", "effectiveness of action", "efforts for action" and "damage of actions", which consider the cost-benefit and side effects of implementing actions. Secondly, it contains a guide to define detailed specific criteria, inferior four general criteria, for each individual context. The idea for specific criteria matches the study objective of the creation of specific indicators.





The definition of specific criteria requires an investigation for every components (technic, social, or organisational) in studied infrastructural systems, their functions and the efforts depend in four dimensions, functional, environmental, economic and human/material resource. The specific criteria definition are particular, adapted to practical situation, and meets the commands of managers. The significance of different criteria may vary between different contexts and scales. For 200 obtaining the specific criteria corresponding to four general criteria suitable, this framework requires the description of four factors in the analysed scenario (in which a target CI is affected by a hazard), including (Fig. 6):

- The "affected system" is therefore a target CI. The resilience of a CI relates to its expected function or services derived by the system of this CI.

- The "hazard" is one or more potential catastrophic events causing negative effects to the target CI, in particular to its 205 function and service;

- The "consequence" refers to the damages to the target CI due to the hazard;

- The "action" could be one or several implementing actions for improving the resilience of the target CI.

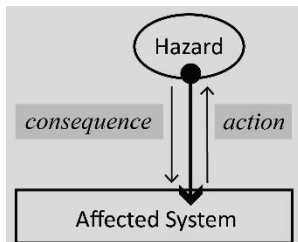

**Fig. 6. Conceptual scenario of resilience, source: Yang et al. (2021).**

**3.1.1 Direct and indirect damages**

The identification of significant damages is based on the use of 'Form 1' (Fig. 7), which is a process to carry out the specific sub-criteria for the criteria "damages to internal components". The last general criterion "Damage of action" requires defining a continuous scenario, in which the optimisation action implemented in the initial scenario causes a side effect to a system connected to the target CI. The sub-criteria corresponding to "damage of action" could be also defined through 215 "Form 1".



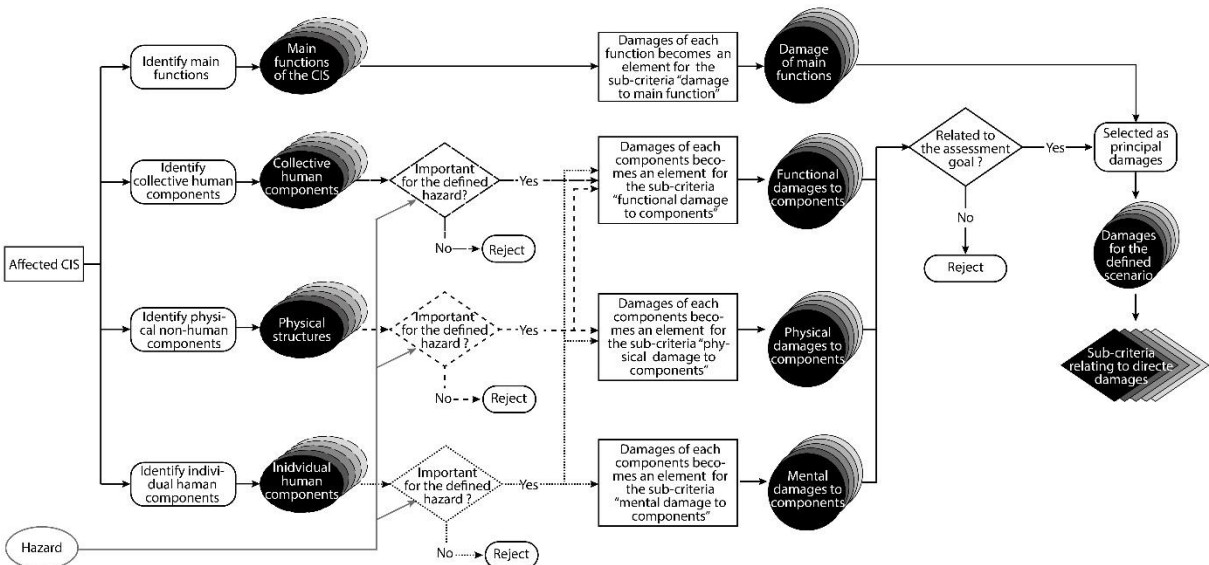

**Fig. 7. Form 1 for defining damage-related sub-criteria "Damage to internal components" and "damage of action", source: Yang et al. (2023).**

### 3.1.2 Effectiveness and damage of actions

For the sub-criteria related to the "action' aspect, before defining sub-criteria, the implementing actions need to be determined. The choice of implementation actions is based on the "Behind the Barriers" model (BB model), developed by Barroca and Serre (2013), which allows effective and comprehensive development of infrastructural system resilience by considering the interdependencies in various urban scales. The applications of the "Behind the Barriers" model to actions identification have been presented in several studies (Gonzva, 2017; Gonzva et Barroca, 2017; Yang et al., 2022; Barroca et

al., 2023). BB model argues that the actions for improving capabilities could be described in four dimensions:

1) A cognitive dimension refers to knowledge, awareness, and the identification of resilience by the persons concerned;

2) A functional dimension specific to material objects and technical urban systems forming the territory;

3) A correlative dimension that recognizes that service and utilisation form a whole whose different sections are interconnected together;

4) An organisational dimension that raises the question of the persons involved (public and private players, populations, etc.) and the strategies that contribute to improving resilience

Then, both implementing actions and the ideal outcome and costs of actions need to be clarified. The process is presented in 'Form 2' (Fig. 8).





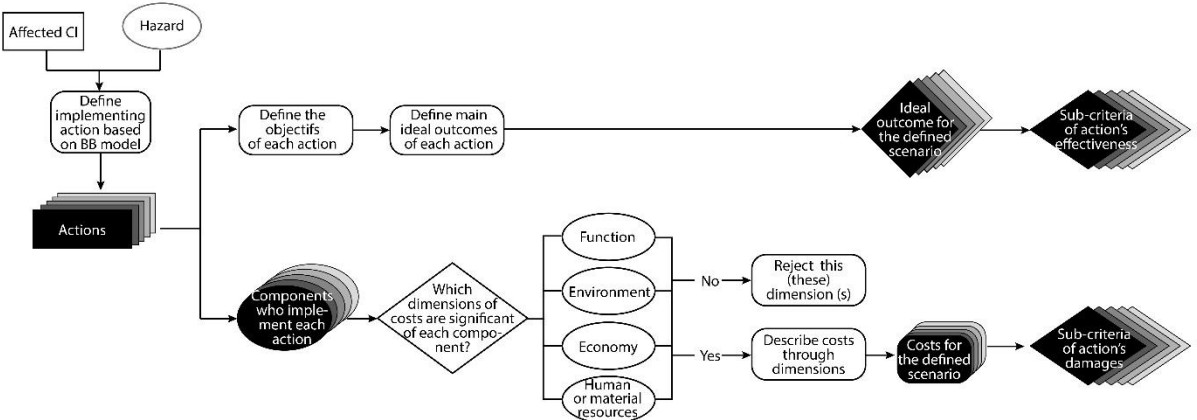

**Fig. 8. Form 2 for defining sub-criteria of "Effectiveness of action" and "Effort for action", source: Yang et al. (2023).**

### 3.2 Creation of Potential Indicators

Each indicator necessities correspond to a criterion. Useful indicators provide information on defined sub-criteria addressing damages, outcomes, and costs. During indicators creation, the existing indicators suitable to defined sub-criteria could be applied directly. For example, for the criteria mentioned – the function of a road network, the indicators presented in the study of Xu and Barth (2006) could be borrowed. It refers to three indicators, Speed, Flow, and Density, whose references involve six levels of service for Basic Freeway Section (see Table 1). However, we emphasise that the adaptability of the criteria depends on the study context. The references in Table 1 are created for Southern California's Inland Empire freeway network; their usefulness on other roads needs adequate proof.

**Table 1. Level of Service for Basic Freeway Section, source: Xu and Barth (2006). Level 1 means a highest function, while level 6 means a lowest function.**

| Grade | Speed Range (mph) | Flow Range (vehicle/hour/lane) | Density Range (vehicle/mile) |
|---|---|---|---|
| 1 | Over 60 | Under 700 Under 12 | Under 700 Under 12 |
| 2 | 57-60 | 700-1100 | 12-20 |
| 3 | 54-57 | 1100-1550 | 20-30 |
| 4 | 46-54 | 1550-1850 | 30-42 |
| 5 | 30-46 | 1850-2000 | 42-67 |
| 6 | Under 30 | Unstable | Above 67 |

On the other hand, indicators and references should be adjusted, modified, and even designed, if no suitable indicators is found in available resources. The indicators for one criterion could be designed by the description through four dimensions:

- The temporal dimension focuses on the duration of the factors.

- The spatial dimension emphasises the spatial or geographical extent of the factors, which can often be represented as a planar or elevation image

- The quantitative dimension relates to the quantifiable data associated with that factors.





- The qualitative dimension relates to non-quantifiable, qualitative data about the factors and might be based on people's
observation and analysis, like the nature (including type, property, characteristic, etc.), importance level, and the degree that needs to be surveyed by experts or operators, such as the indicator "types of vehicles accepted" mentioned in section 2.

For instance, for physical damage of a flooded road infrastructure, possible (not unique) indicators could be pre-defined as:
- Temporal dimension: duration of submersion to road
- Spatial dimension: length of the flooded road
- Quantitative dimension: number of flooded sections
- Qualitative dimension: the importance of flooded sections.

Once possible indicators have been pre-set through four dimensions, references definition for these indicators should be
established. Since the references are extremely pertinent to the object in particular studies, they should rely on the documents, laws, regulations, policies, guidelines, plans, and other information sources provided by relevant institutions or stakeholders in relation to the studied scenario. For instance, for the example qualitative indicator mentioned, "Importance of flooded section", its reference could be set based on the "importance levels of roads" defined by the Institut national de l'information géographique et forestière (IGN). The level of damage caused by flooding increases with the importance of the
road (Table 2). Finding references sometimes requires considering the sources not publicly available. Advantageously, in general, local managers, being part of stakeholders, have a noble knowledge of studied circumstances and could obtain authorisations of critical and non-public information. The final possible indicators should be determined by the content of defined references.

Table 2. Damage reference defined by road importance level, adjusted by IGN (2023).

| Damage level | Importance level of flooded sections | Description |
|---|---|---|
| Catastrophic damage | Importance 1 | The object is of national importance or influence, justifying its representation at scales of 1:1,000,000. |
| Very heavy damage | Importance 2 | The object is of regional importance or influence, justifying its representation at scales of 1:250,000. |
| Heavy damage | Importance 3 | The object is of regional importance or influence, justifying its representation at scales of 1:100,000. |
| Moderate damage | Importance 4 | The object is of inter-communal or cantonal importance or influence, justifying its representation at scales of 1:50,000. |
| Slight damage | Importance 5 | The object is of municipal importance or influence, justifying its representation at scales of 1:25,000. |
| Negligible damage | Importance 6 | The object is of local importance or influence, justifying its representation at scales of 1:5,000. |






## 3.3 Available Data Analysis

Criteria and data are the two structural pillars of indicators creation. Indicators could be measured by historical data or modelling data. Each country exists national databases for different areas and various documents for diverse infrastructures and disasters, which are potential resources for indicators assessments. Three points are emphasised for available data analysis:

- Relevance. The data must be relevant to created indicators and defined criteria. For example, in studying flood disasters, flood-related institutions as well as the data on topics, nature, water resources, disasters, etc., should be the focus of data searching.

- Adaptability. The defined scenario relates certainly to a specific disaster event to which obtained information should be adapted.

- Usability. The long-term availability of the used data ensures continuous assessment. Managers should confirm their authority over obtained data before using them.

Although modern data is diverse (digital database, text, graph, Web, image), since the 1960s databases and information technology have systematically evolved from primitive file processing to complex and powerful database systems. Therefore, if the research involves databases which huge numbers of data, the data mining techniques proposed by Han et al. (2011) are suggested to collect valuable data.

In summary, the process for setting indicators is summarised in Form 3 (Fig. 9). All steps require mutual collaboration of relevant stakeholders or decision-makers, since collaborative approaches ensure the shared diagnosis and the efficiency of implementing measures (Hollnagel et al., 2011).





# Guide of indicators creation for Critical Infrastructures Resilience

Fig. 9. Form 3 for creating indicators, created by authors.

## 4 Guide for Indicators Creation

This study targets a specific occurrence, in which Nantes Ring Road (NRR) is affected by urban flooding, as a scenario for
presenting indicators creation through the designed guide. With a length of 42 kilometres, the NRR has services extending



beyond the local level and is attractive in the region and even in the nation. However, the section (Fig. 10, lines in red) between the "Porte de la Chapelle" (Fig. 10, point B) and the "Porte de la Beaujoire" (Fig. 10, point C) is frequently closed due to the flooding of the Gesvres River. During the closure of this section, local road management DIRO suggests the alternative roads shown in Fig. 10 (lines in green). This case study takes the flood event in February 2020 as an example, when this section was closed on both sides for 56h (Cerema, 2023). Therefore, the first studied scenario refers to the NRR affected by this flood, for which DIRO suggests alternative roads when affected sections are closed (Fig. 11, Initial scenario).

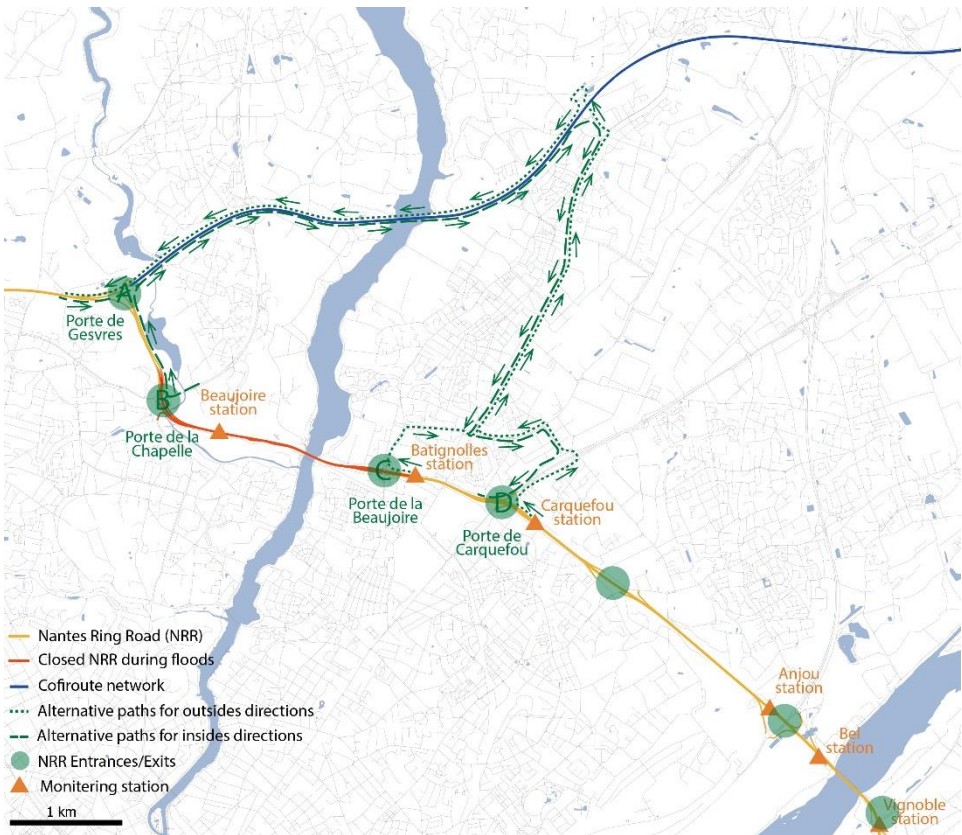

**Fig. 10. Road networks in presented example, adjusted from Cerema (2023).**

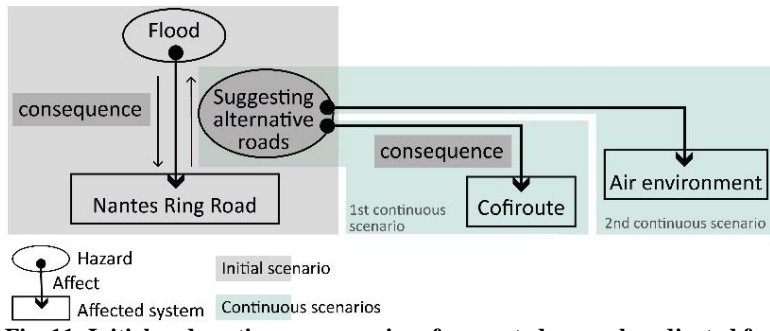

**Fig. 11: Initial and continuous scenarios of presented example, adjusted from Yang et al. (2023).**



For studying the side effects of the implemented action (suggestion of alternative roads), it necessitates defining a continuous scenario, in which this action affects a NRR-connected system. In this example, one part of the alternative path, the section

A11 of the Cofiroute network (Fig. 10, lines in blue), is selected as the first affected system in a continuous scenario (Fig. 11, continuous scenario). The increase in traffic on the Cofiroute network due to the closure of NRR could have negative impacts, like cause congestion, noise pollution, etc. (Cerema, 2023).  Therefore, this example investigates also the "Damage of action" of the initial scenario, i.e. the "the "Damage to internal components" in first continuous scenario. Moreover, as the alternative pathways, which are longer than initial pathways, produces more air pollution. The air environment in Nantes is

chosen as the second affected system in another continuous scenario. Since the action in the continuous scenario presents not as a study objective, it will be omitted.

In decision-making process for risks management, the consideration of experts' opinions is undeniable because of their professional knowledge (Merad, 2010). Therefore, during the whole study process, the research team, including university

scientists, researchers in Cerema (Centre for Studies and expertise on risks, the environment, mobility, and development), and the practicing managers DIRO, make collective decisions based on the content of their meeting discussions.

### 4.1 Specific Criteria Definition

According to section 3.1, the main function of target systems, as well as the function of each internal component are indispensable. This example, therefore, identifies this basic information (Table 3) based on local documents and several

existing studies (Yang et al, 2022; Yang et al. 2023) that investigated NRR resilience. In addition, the desirable outcome and costs of selected optimisation action need to be defined. The ideal outcome of implementing the action would be the increased transport function of the alternative routes. The completion of the action relies directly on two relevant internal components, the "Managers" that plan it and the "Individual users" who use it. The material and economic costs are then dominated by their costs.

**Table 3. All components of NRR and their principal functions, source: Yang et al. (2022).**

| Categories | Internal components | Principal Functions |
|---|---|---|
| Human collective components | Managers | Ensure the daily operation of NRR, providing comfort and safety to users, through the management and maintenance of roads |
| | Project managers | Project management of investment operations (public or private) and management of the noise observatory of the NRR and of the flood-warning project for the eastern part of Highway Infrastructure |
| | State partners | Define and fund projects |
| | Safety observation | Produce and disseminate information on road safety |
| | Collective users | Organize mobilisation for different activities (posters, couriers, travellers, merchandise, health emergency services, etc.) |
| Human individual components | Individual users | Mobilise different activities (posters, couriers, travellers, merchandise, health emergency services, etc.) |
| | Individual staff | Work for affiliated institutions to ensure system functions |
| Physical | Rest areas | Supply energy and fuel to vehicles and provide material and spiritual needs to users in dedicated service |





| structures | | areas |
|---|---|---|
| | Counting regulation | Provide information on road traffic |
| | Access regulation | Improve traffic flow on the Highway Infrastructure by controlling the injection of vehicles |
| | Green spaces | Protect water resources and enhance ecological transparency |
| | Maintenance and intervention centre | Provide support to state institutions (such as the police), cleaning, ordinary and extraordinary maintenance (road signs, lighting, localised damage, etc.) |
| | Drainage system | Remove surface water from the roads as quickly as possible (drainage) to ensure safety with minimum nuisance to users, implement effective subsurface drainage to maximise the lifecycle of infrastructures, minimise the impact of run-off on the external environment in terms of flood risk and water quality |
| | Road structures | Enable mobility by the construction of horizontal structures or structures in elevation or in excavation |
| | Vehicles | Transport passengers and goods on the ground |
| Main Functions | Transport function | Serve individual and collective users in mobility: passenger, freight, postal, or auxiliary transport services (including medical services) |

Based on "form 1" (Fig. 7) and "form 2" (Fig. 8), all sub-criteria for three-defined scenarios are listed in Table 4. The "Damage to transport function", for local manager, is the only significant damage to the the Cofiroute network caused by the implementation of alternative roads. In addition, as the air environment is not an infrastructure system, Table 1 (Fig.7)

therefore does not serve for damage definition. "Damage to the air environment in Nantes" refers in particular to the additional air pollution caused by the increased travel distances via alternative roads.

**Table 4. Sub-criteria defined through "step 1" in "form 3", resulting from consensus of stakeholders and managers.**

| Scenario | Criteria | Sub-criteria |
|---|---|---|
| Initial scenario | Damage to internal components (See Table 3) of NRR | Damage to transport function |
| | | Physical damage to individual users |
| | | Physical damage to road structures |
| | Effectiveness of action | Increased transport function to alternative roads |
| | Effort for action | Resources costs to individual users |
| Initial and continuous scenarios | Damage of action = Damage to internal components (See Table 3) of Cofiroute network | Functional damage of transport function |
| | Damage of action = Damage to air environment in Nantes | Air pollution damage |

**4.2 Possible Indicators and Reference Definition**

In this example, few available indicators could be used directly use for the defined sub-criteria. Thus, we can only follow the steps given in "Form 3" to create indicators and define references. After significance analysis and before reference definition, 17 possible indicators through 4 suggested dimensions could be pre-set (Table 5).

**Table 5. Possible indicators pre-set through the "step 2.1" in 'form 3', resulting from consensus of stakeholders and managers.**

| Criteria | Sub-criteria | Possible indicators (pre-set) | | | | |
|---|---|---|---|---|---|---|
| | | Temporal | Spatial | Quantitative | Qualitative | |
| Damage to internal components (of NRR) | Damage to the transport function of NRR | Duration for unavailable functions | Length of road sections concerning unavailable functions | Reduced transport traffic number | Quality change of transport function | Type of roads sections losing functions |
| | Physical damage to individual users | No significant | No significant | Number of injured or killed passengers | Injury types | |
| | Physical damage to vehicles | No significant | No significant | Number of destroyed vehicles | No significant | |



| | Physical damage to road structures | Duration of destruction of physical structures | Size, scale or length of destroyed physical structures | No significant | Damage level of destroyed physical structures |
|---|---|---|---|---|---|
| Effectiveness of action | Increased transport function of alternative roads | No significant | No significant | Restored traffic | No significant |
| Efforts for action | Resources cost of individual users | Time costs of individual users | No significant | No significant | No significant |
| Damage of action | Functional damage of transport function of Cofiroute Network | Duration for domino effects | Length of road sections concerning domino effects | Reduced transport traffic number | Quality change of transport function |
| | Air pollution in air environment | No significant | No significant | Quantity of pollutant emissions | No significant |


Then, after reviewing a large number of documents published by institutions related to flood management and road infrastructure (Appendix A), 5 of the 17 indicators mentioned above are excluded because their references could not be defined. The description, reference, and resources of the remaining 12 indicators are listed in Tables 6, 7, 8, and 9.

In this example, the difference between the two indicators, "Duration of the NRR close because of submersion" and "Traffic state on the alternative roads" needs to be further explained. They are both used for describing damage to transport function. The traffic flow is more indicative of the damage of the former as it relates to closed road sections. Meanwhile, the latter relates to alternative roads, which have certainly increased traffic flow but with inevitable problems of traffic quality (over-density, congestion, traffic noise, etc.), so that it should be indicative through the traffic state. It shows that the definition of

indicators must be contextualised.

**Table 6. Possible indicators determined through the "step 2.2" in 'form 3' for the criterion "damage to internal components", resulting from consensus of stakeholders and managers.**

| Sub-criterion | Possible indicators | | Reference | Damage score | Reference source | Description in original source |
|---|---|---|---|---|---|---|
| | Pre-set | Determined | | | | |
| Damage to transport function | Duration of destruction of physical structures | Duration of the NRR close | No close | 0 | CGDD (2017) | Damage to the Var bridge and its consequences:<br>Minor damage intensity: 3 days expected outage;<br>Moderate damage intensity: less than 3 weeks planned outage;<br>Moderate damage intensity: less than 3 weeks planned outage;<br>Major damage intensity: less than 3 months expected outage; |
| | | | Close less than 3 days | 1 | | |
| | | | Close between 3 and 30 days | 2 | | |
| | | | Close between 30 and 120 days | 3 | | |
| | | | Close between 120 days and 2 years | 4 | | |
| | Quality change of transport function | Traffic flow on the affected NRR sections | Flow > 100 vehicles/6minutes | 0 | Cerema (2023) | Characterisation of road transport operation by flow rate:<br>Flow > 100 vehicles/6minutes = high flow<br>Flow between 50 and 100 vehicles/6 minutes = moderate flow<br>Flow < 50 vehicles/6 minutes = low flow |
| | | | Flow between 50 and 100 vehicles/6 minutes | 1 | | |
| | | | Flow < 50 vehicles/6 minutes | 2 | | |
| | Type of roads sections losing functions | Importance of closed road sections | No flooded road structures | 0 | IGN (2023) | See section 3.2 |
| | | | Importance level 6 | 1 | | |
| | | | Importance level 5 | 2 | | |
| | | | Importance level 4 | 3 | | |
| | | | Importance level 3 | 4 | | |
| | | | Importance level 2 | 6 | | |





| | | | Importance level 1 | 7 | | |
|---|---|---|---|---|---|---|
| Physical damage to individual users | Number of injured or killed passangers | Number of injured users | No injured passenger | 0 | SETRA (2005) | ZAAC (zone d'accumulation d'accidents corporels) is defiend through the number of accident for a road section length of 850 m and over a period of 5 years: Level 1: at least 4 accidents with injuries and 4 serious casualties; Level 2: at least 7 accidents with injuries and 7 serious casualties Level 3: at least 10 accidents with injuries and 10 serious casualties |
| | | | 4 injured passenger for each 850m | 1 | | |
| | | | 7 injured passenger for each 850m | 2 | | |
| | | | 10 injured passenger for each 850m | 3 | | |
| | | Number of killed users | No dead | 0 | Defossez, (2009) | Human damage severity scale |
| | | | 1 à 9 dead | 1 | | |
| | | | More than 9 dead | 2 | | |
| | Injury types | Injury grade of injured passengers | No injured passenger | 0 | Sécurité routière | In-patient casualties: Victims admitted to hospital as patients for more than 24 hours. Minor injuries: Victims who have received medical care but have not been admitted to hospital as in hospital for more than 24 hours. |
| | | | slightly injured | 1 | | |
| | | | Serious injured | 2 | | |
| Physical damage to road structures | Duration of destruction of physical structures | Duration of NRR flooding | 0 | 0 | Cerema (2016) | The duration of submersion could be classified as: less than 24 h-1 day from 24 to 48 h from 48 h-2 days to 4 days from 5 to 10 days more than 10 days |
| | | | Less than 24 h | 1 | | |
| | | | 24 h - 48 h | 2 | | |
| | | | 2 – 4  days | 3 | | |
| | | | 5-10 days | 4 | | |
| | | | More than 10 days | 5 | | |
| | Damage level of destroyed physical structures | Percentage of Pavement Damage | Insignificant Damage | 0 | Lu (2019) | Damage states can be categorized based on damage level such as collapse, major damage, moderate damage and minor damage, according to the percentage of pavement damage |
| | | | Minor Damage | 1 | | |
| | | | Medium Damage | 2 | | |
| | | | Minor Damage | 3 | | |
| | | | Insignificant Damage | 4 | | |

**Table 7. Possible indicators determined through the "step 2.2" in 'form 3' for the criterion "effectiveness of action", resulting from consensus of stakeholders and managers.**

| Sub-criterion | Possible indicators | | Reference | Recovery score | Reference sources | Description in original source |
|---|---|---|---|---|---|---|
| | Pre-set | Determined | | | | |
| Increased transport function of alternative roads | Restored traffic | Percentage of traffic being restored on alternative roads* | 0 | 0 | No available | Non available |
| | | | 0-30% | 1 | | |
| | | | 30%-60% | 2 | | |
| | | | More than 60% | 3 | | |

*This indicator is intended to indicate the percentage of vehicles affected and using the alternative route to the total number of vehicles affected

**Table 8. Possible indicators determined through the "step 2.2" in 'form 3' for the criterion "effort for action", resulting from consensus of stakeholders and managers.**

| Sub-criterion | Possible indicators | | Reference | Cost score | Reference sources | Description in original source |
|---|---|---|---|---|---|---|
| | Pre-set | Determined | | | | |
| Resources costs of individual users | Time costs of individual users | Additional time depend by each user in using alternative road | Less than 15 minutes | 1 | BFM business (2018) | 82% of French people lose patience after 30 minutes of non-fluid driving, and 40% after just 15 minutes. |
| | | | 15-30 minutes | 2 | | |
| | | | More than 30 minutes | 3 | | |

**Table 9. Possible indicators determined through the "step 2.2" in 'form 3' for the criterion "damage of action", resulting from consensus of stakeholders and managers.**

| Sub-criteria | Possible indicators | | Reference | Damage score | Reference sources | Description in original source |
|---|---|---|---|---|---|---|
| | Pre-set | Determined | | | | |
| Damage to | Quality | Traffic state on | fluid | 0 | Nantes | Nnates metropole has defined the traffic situation as follows: |





| transport function of Cofiroute Network | change of transport function | the alternative roads | saturated | 1 | metropole | Lane occupancy rate less than 20%: Fluid<br>Lane occupancy between 20% and 30%: Dense<br>Occupancy rate between 30% and 40%: Saturated<br>Lane occupancy rate above 40%: Blocked |
| | | | dense | 2 | | |
| | | | blocked | 3 | | |
| Air pollution in air environnement | Quantity of additional pollutant emissions | Percentage of additional CO2 emission for each path through alternative road | 0-93% | 1 | phys.org | Due to 380 billion tons of CO₂ as the remaining carbon budget, there is a 50% chance the planet will reach the 1.5°C global average temperature rise in just nine years. when the remaining carbon budget increases 93% to 732 billion tons or 224% to 1230 billion tons, the global average value of temperature rise could become 1.5°C and 2°C. |
| | | | 93-224% | 2 | | |
| | | | more 224% | 3 | | |

**4.3 Data Collection**

The example relates to flood hazards and road infrastructures. In France, relevant institutions are presented Appendix A, while the possible data resources could be found in the open data websites shown in Appendix B. Moreover, the partner DIRO provides a large number of data on traffic flow on the NRR network. The indicator "Percentage of Pavement Damage" is rejected due to lack of data. All serviceable indicators and their suitable data resources are listed in Table 10. The main data used sources refer to:

- The traffic flow per six minutes monitored by 18 channels in four stations on NRR, collected by DIRO: four channels in four stations (Beaujoire, Batignolles, Carquefou, and Vignoble) for both two directions, whereas Anjou station has only two channels for the internal direction. Collected data are relevant to two periods: 1) the first is from 14 to 20 January 2019 and is considered a normal situation; and 2) the second is from 31 January to 07 February 2020 and is considered a flooding situation. 62 676 data of traffic flow are involved.
- The BDTOPO from IGN on the department of Loire-Atlantique, which includes a 3D vector description (structured in objects) of road infrastructures.
- Documents from relevant local institutions, like DIRO, Cerema, Nantes Metropole.

**Table 10. Data resources selected through the "step 3" in 'form 3', resulting from consensus of stakeholders and managers.**

| N° | Indicators | Data resources |
| --- | --- | --- |
| 1 | Duration of the NRR close | DIRO |
| 2 | Traffic flow on the affected NRR sections | DIRO |
| 3 | Importance of closed road sections | IGN |
| 4 | Number of injured users | Local news |
| 5 | Number of killed users | Local news |
| 6 | Injury grade of injured passengers | Local news |
| 7 | Duration of NRR flooding | DIRO |
| 8 | Percentage of traffic being restored on alternative roads | DIRO |
| 9 | Additional time costs | IGN |
| 10 | Additional co2 emission | IGN |
| 11 | Traffic state on the alternative roads | Nantes metropole |

# 5 Resilience Assessment

As shown in Fig. 1, resilience could be assessed through Criteria & Indicators, and the latter could be assessed by reliable data. After criteria definition, indicators creation, and data selection, the assessment process for the resilience of the studied



CI, including potentially 5 phases (Fig. 12), could be completed. it should determination of assessment methods and
weighting methods. As numerous methods are deployable, this example shows only some of them considered applicable and
suitable for defined criteria and created indicators.

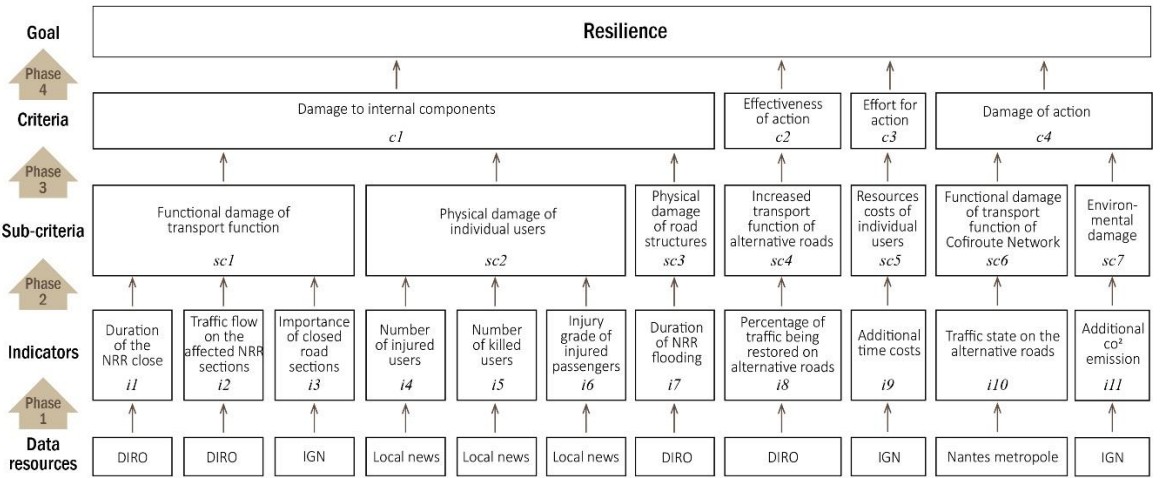

**Fig. 12. Assessment process of studied example based on defined criteria, created in dictators and collected data, created by authors.**

## 5.1 Criteria & Indicators Weighting

The developed guide, inspired by criteria & indicators setting analysis, could be considered as a multi-criteria study. Multi-criteria evaluation is a very efficient tool to implement a multi/inter-disciplinary approach and many scientists have demonstrated its usefulness in many sustainability options and management problems. MCDM is a branch of operational research dealing with finding ideal results in complex scenarios including various indicators, conflicting objectives, and

criteria (Kumar et al., 2017). A multi-criteria framework investigation makes the assessment applicable in practice by managers, and makes it possible to consider different alternatives and the multidimensionality of the real world, to address different realities in the infrastructure assessment (Sierra et al., 2018). Since MCDM requires consideration of various perspectives, weighing methods is regarded as an important aspect in the MCDM methods step as the results of the multi-criteria decision-making method largely depend on such weights (Yusop et al., 2015). Weighting values accurately

determine the relative importance of each factor significant to assessments (Singh and Pant, 2021). Even though most of MCDM studies highlight the weighting of criteria, this study considers its utilisation for all criteria and indicators. The weighting process in the MCDM approach is the most difficult task (Tervonen et al., 2009), even though weighting methods have been popular in recent years. A significant scientific system has therefore been developed and there are many available methods presented in a large number of studies. The relevant review articles are listed here and this study would not present

in detail: Roszkowska, 2013; Johnsen and Løkke, 2013; Iwaro et al., 2014; Yusop et al., 2015; Singh and Pant, 2021.





Weighting methods could be simply divided into two categories, Subjective Weighting Methods, and Objective Weighting Methods. The former involves weights being derived from the decision maker's judgment, while the latter preference weights are obtained from mathematical algorithms or models (Yusop et al., 2015). Subjective Weighting Methods are more
suitable for the example that aims at helping each decision maker to implement assessments according to specific requirements and judgments based on particular situations. Moreover, the present study selects the weighting methods that do not require additional software, and that do not require excessive simulation or mathematical skills that are difficult to be applied by managers in practice. The existing methods are numerous and it is difficult to show all of them. This section will use different methods to assess criteria level and indicators for presenting some of the existing methods. All methods
mentioned following are based on the study of (Yusop et al., 2015).

For the sub-criteria with only one indicator (indicators 7, 8, 9, 10, 11), indicators weighting is not necessary. For the resting sub-criteria, several weighting methods widely used for a small number of elements are suggested, as there are no more than three indicators for each sub-criterion in the example. Firstly, the ranking methods, such as rank sum and rank reciprocal, are
the simplest approach for assigning weights. Generally, before calculating weights, the criteria are ranked in order from most important to least important. "In rank sum, the rank position rj is weighted and then normalized by the sum of all weights. Rank reciprocal weights are derived from the normalized reciprocals of a criterion rank. The rank exponent method requires the decision maker to specify the weight of the most important element on a 0–1 scale. The value is then used in a numerical formula." Yusop et al. (2015). The results of indicators weighting shown in Table 11.


**Table 11. Indicators weights, created by authors.**

| Sub-criteria | N° | Indicators | | Straight rank | Rank sum $(n - r_j + 1)$ | |
|---|---|---|---|---|---|---|
| | | | | | Weight | Normalised |
| Functional damage of transport function | 1 | *i1* | Duration of destruction of physical structures | 2 | 2 | 0.33 |
| | 2 | *i2* | Quality change of transport function | 1 | 3 | 0.50 |
| | 3 | *i3* | Importance of closed road structures | 3 | 1 | 0.17 |
| | | | | | 6 | 1 |
| Physical damage of individual users | 4 | *i4* | Number of injured users | 2 | 2 | 0.33 |
| | 5 | *i5* | Number of killed users | 1 | 3 | 0.50 |
| | 6 | *i6* | Injury grade of injured passengers | 3 | 1 | 0.17 |
| | | | | | 6 | 1 |

Ranking methods are not ideal for weighting no more than two elements, as only two ranges are taken into account. Therefore, for criteria and sub-criteria weighting, another easy weighting method called the Point Allocation method, could
be suggested. "In the point allocation weighting method, the decision maker allocates numbers to describe directly the weights of each criterion. The decision maker is asked, for example, to divide 100 points among the criteria. In many experiments, the analysts do not fix the total number of points to be divided but the subjects are asked to give any numbers they liked to reflect the weights. The more points a criterion receives, the greater its relative importance. The total of all criterion weights must sum to 100." Yusop et al. (2015). Similarly, for the criteria with only one sub-criterion, weighting is
not necessary. The results of sub-criteria weighting are shown in Table 12.



**Table 12. Sub-criteria weights, created by authors.**

| | Sub-criteria | Rank sum | |
|---|---|---|---|
| | | Weight | Normalized |
| *wsc1* | Damage to transport function | 30 | 0.3 |
| *wsc2* | Physical damage to individual users | 50 | 0.2 |
| *wsc3* | Physical damage to road structures | 20 | 0.2 |
| | | 100 | 1 |
| *Wsc6* | Functional damage to transport function of Cofiroute Network | 80 | 0.8 |
| *Wsc7* | Air pollution in air environment | 20 | 0.2 |
| | | 100 | 1 |

For criteria weighting, this study suggests Direct Rating Method. This method requires a score, like the numbers 1–5, 1–7, or

1–10 used to indicate importance from a decision maker to represent the importance of each indicator. Yusop et al. (2015 ) argued that "The rating method does not constrain the decision maker's responses as the fixed point scoring method does. It is possible to alter the importance of one criterion without adjusting the weight of another. This represents an important difference between the two approaches." Thus, the results of criteria weighting are shown in Table 13.

**Table 13. Criteria weights, created by authors.**

| N° | Criteria | Importance (1 = least, 5 = most) | | | | | Level | Normalised weight | |
|---|---|---|---|---|---|---|---|---|---|
| | | 1 | 2 | 3 | 4 | 5 | | | |
| 1 | Damage to internal components | | | | X | | 4 | *wc1* | 0.308 |
| 2 | Performance of action | | | | | X | 5 | *wc2* | 0.384 |
| 3 | Efforts of action | | X | | | | 2 | *wc3* | 0.154 |
| 4 | Damage of action | | X | | | | 2 | *wc4* | 0.154 |
| | | | | | | | 13 | | 1 |

**5.2 Assessment Methods and Results**

Assessment could be quantitative, qualitative and semi-quantitative (Yang et al., 2023-b). Quantitative approaches offer domain-agnostic measures to quantify value across applications and structural-based modelling approaches that model domain-specific representations. Semi-quantitative approaches provide a general numerical description of the classification, without detailed formulae or models. Qualitative approaches refer to approaches without a numerical descriptor and based on

people's judgments and analysis, like surveyed experts or operators (Hosseini, 2016; Cantelmi et al., 2021).

The hierarchical references of created indicators suggested in this study make indicators assessment a semi-quantitative approach (Fig. 12. Phase 1). Based on the collected data, all indicators could be assessed. The values and levels of all indicators for the defined scenario are assessed below.

**5.2.1. Indicators Assessment**

**Indicator 1 - Duration of the NRR close**
According to an internal document of Cerema (2023), in February 2020, the maximum height of the Gesvres at the Jonelière station reached 251 cm and traffic were closed with a disruption at lasted 56h.

**Indicator 2 - Traffic flow on the affected NRR sections**





Four monitoring stations and their 14 channels are involved in the affected section, Batignolles, Carquefou, Anjou, and Vignoble. The weights of the data monitored by 14 channels are calculated by the rank sum method and based on their distance ranking from the affected road and their average traffic flow: the channel closer to the affected section has a higher weight; the channel relating to more traffic flow has a higher weight. The selected data are relating to the traffic flow
between 7 am to 9 am (2 h) on Monday 3 February 2020 (flooding situation) and Monday 14 January 2019 (normal situation). These data have been selected mainly due to the limitations of the data available and their significance. They allow to make comparisons between flooding and normal conditions on the same day of the week. The average traffic flow of the relevant four monitoring stations is shown in Table 14.

**Table 14. Average traffic flow in normal and flooding situations, created by authors.**

| | | | Average flow in Normal situation | Straight rank | Weight | Average flow in Flooding situation |
|---|---|---|---|---|---|---|
| Vignoble | Inside direction | Channel 1 | 123.81 | 13 | 0.02 | 99.62 |
| | | Channel 2 | 108.43 | 14 | 0.01 | 82.86 |
| | Outside direction | Channel 3 | 144.67 | 12 | 0.03 | 31.24 |
| | | Channel 4 | 200.67 | 11 | 0.04 | 169.33 |
| Anjou | Inside direction | Channel 1 | 62.71 | 10 | 0.05 | 45.15 |
| | | Channel 2 | 135.52 | 9 | 0.06 | 59.43 |
| Carquefou | Inside direction | Channel 1 | 113.29 | 5 | 0.10 | 23.29 |
| | | Channel 2 | 83.81 | 7 | 0.08 | 4.10 |
| | Outside direction | Channel 3 | 79.14 | 8 | 0.07 | 0.00 |
| | | Channel 4 | 95.14 | 6 | 0.09 | 0.00 |
| Batignolles | Inside direction | Channel 1 | 132.71 | 1 | 0.14 | 0.00 |
| | | Channel 2 | 111.52 | 2 | 0.13 | 0.00 |
| | Outside direction | Channel 3 | 78.10 | 4 | 0.11 | 0.00 |
| | | Channel 4 | 97.42 | 3 | 0.12 | 0.00 |
| Average | | | 109.52 | | | 19.01 |

**Indicator 3 - Importance of closed road sections**

According to the BDTOPO of the department of Loire-Atlantique, the closed section has 29 parts, of which twenty are categorised as importance level 1, seven are categorised as importance level 3, and two are categorised as importance level 5.
Consequently, the value of average importance is 1.76.

**Indicator 4, 5, 6 - Number of injured users, Number of killed users, Injury grade of injured passengers**

According to the local document that descript the studied flooding event, no injured, dead, or destroyed vehicles were caused by this flood event.


**Indicator 7 - Duration of NRR flooding**

According to Cerema (2023), NRR is inundated for 60 h (Figure 13). The duration of the NRR being flooded differs from the duration of the NRR being closed because roads do not need to be closed if the flooding does not affect the traffic function. The duration of the NRR being flooded is about the physical damage to road infrastructure, while the duration of
the NRR being closed is related to the functional damage to road infrastructures.



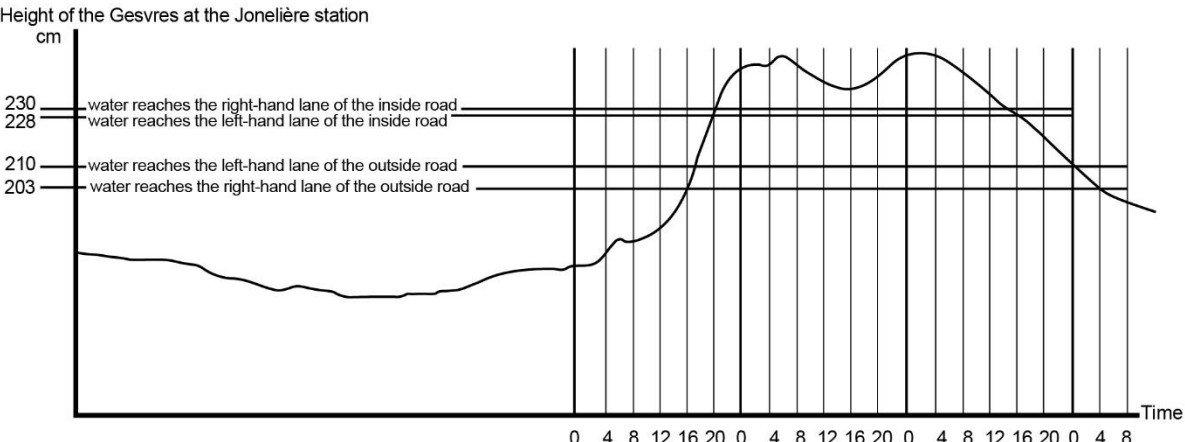

**Fig. 13. Duration of the NRR (inside and outside roads) being flooded.**

**Indicator 8 - Percentage of traffic being restored on alternative roads**

The closed section shows in Fig. 10. According to Cerema (2023), there are 4800 passages increased on the alternative path of inside direction on Sunday 2 February 2020 between 6 pm to 7 pm (1h). Therefore, the selected data are relating to the traffics on NRR between 6 pm to 7 pm on Sunday 2 February 2020 (flooding situation) and Sunday 20 January 2019 (normal situation). Because of the road closure, traffic on all four monitoring stations is affected and their traffic flow is shown in

Table 15 below. It can be seen that the closer the road is to the affected section, the more it is affected. 5 166 passengers were lost in one hour in the inside direction of the NRR, of which 4 800 are received by the alternative path.

**Table 15. Total traffic number in normal and flooding situations, created by authors.**

| Station | Direction | Channel | Total traffic in Normal situation | Total traffics in Flooding situation | Additional traffic on alternative road during closure |
|---|---|---|---|---|---|
| Batignolles | Inside direction | Channel 3 | 1 360 | 1 | 4 800 |
| | | Channel 4 | 667 | 0 | |
| Carquefou | Inside direction | Channel 3 | 1 276 | 213 | Reduced traffic on NRR during closure |
| | | Channel 4 | 630 | 25 | |
| Anjou | Inside direction | Channel 1 | 1 217 | 551 | 5 166 |
| | | Channel 2 | 728 | 279 | |
| Vignoble | Inside direction | Channel 3 | 1 142 | 963 | Percentage of traffic being restored |
| | | Channel 4 | 647 | 469 | 92.92% |
| Total | | | 7 667 | 2 501 | |

**Indicator 9 and 11 - Additional time costs, Additional co2 emission**

Based on the study of Yang et al. (2021), additional travel time and additional $CO_2$ emissions for each vehicle that pass the four alternative roads are shown in Table 16. Moreover, this study adds the weight of each path is based on the total traffic of the original paths of both sides, in a normal situation. For example, a normal situation refers to 15 h 48 (0 am to 3.48 pm) on Monday 14 January 2019, which corresponds to a flooding situation, 15 h 48 (0 am to 3.48 pm) on Monday 2 February 2020.

Thus, the weight of External direction (E) or Internal direction (I) may be defined as Eq. (1):





$$w\,(E,I) = \frac{T(E,I)}{TT}/2$$

"T" is the traffic of internal or external directions. "TT" is the total traffic of both two directions. Consequently, the average

additional travel time is 6 min 5 s and average growth rate of $CO_2$ emission is 152%.


**Table 16. Additional travel time and $CO_2$ emission for each alternative path, adjusted from Yang et al. (2021).**
**"F"=flooding situation, "N"=normal situation, "o"=outside direction, "i"=inside**

| Start and arrival point | Paths | Distance (m) | Travel time | $CO_2$ emission (g) | Traffic of two directions in normal situation | Total traffics | Weight |
|---|---|---|---|---|---|---|---|
| Outside direction, from C to A | *No1* | 3 676 | 2 min 46 s (166 s) | 610 | Traffic of external direction : T(E) = 17 543 | T=36 261 | 0.243 |
| | *Fo1* | 9 732 | 8 min 17 s (497 s) | 1 615 | | | |
| | / | | 5 min 31 s (331 s) | growth rate : 165% | | | |
| Outside direction, from D to A | *No2* | 4 867 | 3 min 40 s (220 s) | 808 | | | 0.243 |
| | *Fo2* | 10 536 | 9 min 00 s (540 s) | 1 749 | | | |
| | / | | 5 min 20 s (320 s) | growth rate : 116% | | | |
| Inside direction, from B to D | *Ni1* | 3 605 | 2 min 42 s (162 s) | 598 | Traffic of internal direction : T(I) = 18 718 | | 0.258 |
| | *Fi1* | 11 125 | 9 min 50 s (590 s) | 1 847 | | | |
| | / | | 7 min 8 s (428 s) | growth rate : 209% | | | |
| Inside direction, from A to D | *Ni2* | 4 731 | 3 min 32 s (212 s) | 785 | | | 0.258 |
| | *Fi2* | 10 151 | 8 min 53 s (533 s) | 1 685 | | | |
| | / | | 5 min 21 s (321 s) | growth rate : 115% | | | |

**Indicator 10 - Traffic state on the alternative roads**

According to the private document of Crema (2023), during NRR closures, the alternative roads carry too much traffic and

cause congestion, especially during the morning and evening rush hours. Furthermore, level normalisation is necessary for

the indicators with a variable number of reference levels but corresponding to the same criterion (Table 17).

**Table 17. The values, scores and normalised scores of each indicators score, created by authors.**

| N° | Indicators | Reference | Score | Score normalisation | Indicator value | Indicator score | Normalised score of indicator |
|---|---|---|---|---|---|---|---|
| 1 | Duration of the NRR close | No close | 0 | 0 | 56h | 1 | 0.25 |
| | | Close less than 3 days | 1 | 0.25 | | | |
| | | Close between 3 and 30 days | 2 | 0.50 | | | |
| | | Close between 30 and 120 days | 3 | 0.75 | | | |
| | | Close between 120 days and 2 years | 4 | 1 | | | |
| 2 | Traffic flow on the affected NRR sections | Flow > 100 vehicles/6minutes | 0 | 0 | 19.01 | 2 | 1 |
| | | Flow between 50 and 100 vehicles/6 minutes | 1 | 0.5 | | | |
| | | Flow < 50 vehicles/6 minutes | 2 | 1 | | | |
| 3 | Importance of closed road sections | No flooded road structures | 0 | 0 | 1.76 | 5 | 0.83 |
| | | Importance level 6 | 1 | 0.17 | | | |
| | | Importance level 5 | 2 | 0.33 | | | |
| | | Importance level 4 | 3 | 0.5 | | | |
| | | Importance level 3 | 4 | 0.67 | | | |
| | | Importance level 2 | 5 | 0.83 | | | |
| | | Importance level 1 | 6 | 1 | | | |
| 4 | Number of injured users | No injured passenger | 0 | 0 | 0 | 0 | 0 |
| | | 4 injured passenger for each 850m | 1 | 0.33 | | | |
| | | 7 injured passenger for each 850m | 2 | 0.67 | | | |
| | | 10 injured passenger for each 850m | 3 | 1 | | | |





| 5 | Number of killed users | No dead | 0 | 0 | 0 | 0 | 0 |
| | | 1 à 9 dead | 1 | 0.5 | | | |
| | | More than 9 dead | 2 | 1 | | | |
| 6 | Injury grade of injured passengers | No injured passenger | 0 | 0 | 0 | 0 | 0 |
| | | slightly injured | 1 | 0.5 | | | |
| | | Serious injured | 2 | 1 | | | |
| 7 | Duration of NRR flooding | 0 | 0 | 0 | 60 h | 3 | 0.75 |
| | | Less than 24 h | 1 | 0.25 | | | |
| | | 24 h - 48 h | 2 | 0.5 | | | |
| | | 2 – 4 days | 3 | 0.75 | | | |
| | | More than 4 days | 4 | 1 | | | |
| 8 | Percentage of traffic being restored on alternative roads | 0 | 0 | 0 | 92.92% | 3 | 1 |
| | | 0-30% | 1 | 0.33 | | | |
| | | 30% - 60% | 2 | 0.67 | | | |
| | | More than 60% | 3 | 1 | | | |
| 9 | Additional time costs | Less than 15 minutes | 1 | 0.33 | 6 min 5 s | 1 | 0.33 |
| | | 15-30 minutes | 2 | 0.67 | | | |
| | | More than 30 minutes | 3 | 1 | | | |
| 10 | Traffic state on the alternative roads | fluid | 0 | 0 | dense | 2 | 0.67 |
| | | saturated | 1 | 0.33 | | | |
| | | dense | 2 | 0.67 | | | |
| | | blocked | 3 | 1 | | | |
| 11 | Additional co2 emission | 0-93% | 1 | 0.33 | 152% | 2 | 0.67 |
| | | 93-224% | 2 | 0.67 | | | |
| | | more 224% | 3 | 1 | | | |


### 5.2.2. Determination of the levels of Criteria and Sub-criteria

In order to make judgements, the levels of each criterion (and sub-criterion) could be designed to show what the extent of damage, cost and recovery is. Thus, for phases 3 (indicator to sub-criteria) and 2 (sub-criteria to criteria) in Fig. 12, the aggregated score of indicators should correspond to one level of criteria or sub-criteria. For ease of understanding, this study

simply divides the criteria into five levels: 1 (value 0-2); 2 (value 2-4) ; 3 (value 4-6) ; 4 (value 6-8) ; 5 (value 8-10).Moreover, simple overlay operations with weights can be considered, because the sub-criteria and indicators derived from each criterion are part of its field (Table 18).

**Table 18. The levels of sub-criteria and criteria.**

| Indicators | Score | Weight | aggregated score | Sub-criteria | Level (score) | Weight | aggregated score | Criteria | Level (score) |
|---|---|---|---|---|---|---|---|---|---|
| | | | | | | | | | |
| Duration of the NRR close | 0.25 | 0.27 | 0.77 | Functional damage of transport function | 4 (0.77) | 0.3 | 0.38 | Damage to internal components | 2 (0.38) |
| Traffic flow on the affected NRR sections | 1 | 0,55 | | | | | | | |
| Importance of closed road sections | 0.83 | 0.18 | | | | | | | |
| Number of injured users | 0 | 0.27 | 0 | Physical damage of individual users | 0 (0) | 0.2 | | | |
| Number of killed users | 0 | 0.55 | | | | | | | |
| Injury grade of injured passengers | 0 | 0.18 | | | | | | | |
| Duration of NRR flooding | 0.75 | 1 | 0.75 | Physical damage of road structures | 4 (0.75) | 0.2 | | | |
| | | | | | | | | | |
| Percentage of traffic being restored on alternative | 1 | 1 | 1 | Increased transport function of | 5 (1) | 1 | 1 | Effectiveness of action | 5 (1) |



| | | | | | | | | | |
|---|---|---|---|---|---|---|---|---|---|
| roads | | | | alternative roads | | | | | |
| | | | | | | | | | |
| Additional time costs | 0.33 | 1 | 0.33 | Resources costs of individual users | 2 (0.33) | 1 | 0.33 | Efforts for action | 2 (0.33) |
| | | | | | | | | | |
| Traffic state on the alternative roads | 0.67 | 1 | 0.67 | Functional damage of transport function of Cofiroute Network | 4 (0.67) | 0.8 | 0.67 | Damage of action | 4 (0.67) |
| Additional co2 emission | 0.67 | 1 | 0.67 | Air pollution in air environment | 4 (0.67) | 0.2 | | | |

### 5.2.3. Resilience Assessment

Next, the resilience of studied CI could be assessed. Among existing methods, this study highlights a quantitative assessment method "probabilistic framework", created by Mebarki et al. (2012), as an example. This method, originally created for assessing seismic vulnerability, builds mathematical models by analysing the probability of events occurring.

Furthermore, the unified theoretical approach for resilience, developed by Mebarki (2017), allows an engineering analysis for the resilience of any system, as it considers:

- The prior definition of the system, its components and sub-systems, and the expected utility functions or services, which the system should deliver. These functions or services can be described as a vector (case of multiple expected functions) or a scalar value (case of a unique function or service, or a weighted combination of the whole expected utility functions). The utility function, herein, is denoted $R(t)$ as it depends on time.

- The evaluation of the utility function loss, which loss is denoted $D_R$ with values ranging within the interval [0..1], i.e. no damage up to full damage respectively.

- The capacity of the system to recover at post-damage phase, where the recovering function is denoted $\Phi_a$, which depends on the dynamics of the system. Actually, the system can either recover, or go into worse evolution or remain at residual level with no more variation. This recovery function should be modelled by the physical behaviour or response of the system after some actions are provided.

- This recovery capacity (or worsening function) is also affected by the prior existence of available resources at internal level (within the system) or at external level (through interaction from outside the system). As it is a conditional aspect, it's described by a probabilistic parameter denoted $\chi_r$ which is described as the combination of external or internal resources i.e. split up into two parts $\chi_{m,r}^{int}$ and $\chi_{m,r}^{ext}$.

- The capacity to manage the post-damage phase which capacity is described by a probabilistic parameter, denoted $\chi_{m,c}$.

In the present paper, the authors will consider the post-damage phase and will describe the effects of the adaptive options, which will influence therefore the recovery function $\Phi_a$.

These adaptive options will be discussed in the present paper under various aspects:





-     The efficiency of these actions in terms of recovery function

     -     The availability of the resources in order to set up these actions

     -     The secondary effects of these actions, their consequences on damage amplification as well as the cost for their setup and the expected cost of their secondary and side effects.

     -     The satisfaction of the multi stakeholders that are concerned by the system and its expected utility functions.

**Stakeholders and global satisfaction**

Since various adaptive options can be setup, it's important to investigate their global cost as well as their efficiency, besides the satisfaction of the stakeholders. In fact, this satisfaction can be very subjective. However, there is also an objective way to quantify this satisfaction through statistics.

We propose then the following modelling Eq. (2):

$E_{sh\_satisfaction} = E_{pa} \cap \bar{E}_{da}$

Where:

     -     $E_{SH\_satisfaction}$ = event for which the stakeholders are satisfied, with probability of occurrence denoted $P(E_{SH\_satisfaction})$

     -     $E_{pa}$= Event of efficient action against the first hazard, with probability of occurrence denoted $P(E_{pa})$

     -     $E_{da}$= Event of damaging side effect of first action, with probability of occurrence denoted $P(E_{da})$. The complementary event is denoted $\bar{E}_{da}$, i.e. it is related to non-damaging side effects.


So that the probability of satisfaction can be written as Eq. (3):

$$P\left(E_{sh_{satisfaction}}\right) = P(E_{pa}).\,P(\{\bar{E}_{da}|E_{pa}\}) \xrightarrow{yields} P\left(E_{sh_{satisfaction}}\right) = P(E_{pa}).\,(1 - P(\{E_{da}|E_{pa}\}))$$

With Eq. (4) and Eq. (5):

$$P(E_{Pa}) = P(E_{availabilityOfRequiredResources}).\,P(\{E_{pa}|E_{availabilityOfRequiredResources}\})$$

$$\xrightarrow{yields} P(E_{Pa}) = \begin{cases} 0 : if \begin{cases} P(E_{availabilityOfRequiredResources}) = 0 \,.\,or. \\ P(\{E_{da}|E_{availabilityOfRequiredResources}\}) = 1 \end{cases} \\ 1 : if \begin{cases} P(E_{availabilityOfRequiredResources}) = 1 \,.\,and. \\ (\{E_{da}|E_{availabilityOfRequiredResources}\}) = 0 \end{cases} \end{cases}$$

***Remark***: The limit cases for which the stakeholder has 0 or 1 as satisfaction probability correspond to Eq. (6):

$$P\left(E_{sh_{satisfaction}}\right) = \begin{cases} 0 : if \begin{cases} P(E_{pa}) = 0 \,.\,or. \\ P(\{E_{da}|E_{pa}\}) = 1 \end{cases} \\ 1 : if \begin{cases} P(E_{pa}) = 1 \,.\,and. \\ P(\{E_{da}|E_{pa}\}) = 0 \end{cases} \end{cases}$$

The advantage of such description thanks to probabilistic modelling is that the whole parameters are objective to which are assigned metrics. These metrics, probabilities herein, are obtained by either theoretical distribution modelling or by inquiries.



**Global cost and decision-making**

Targeting resilience supposes that, as described hereabove, several adaptive options, at the post-disaster stage, or the risk
reduction options and preparedness, before any disaster occurs, can be set up. These options suppose that resources are available, are well managed and that their cost are acceptable.

It is then crucial to define the global cost on which will rely the decision-making. For such global cost, we propose the following Eq. (7):

$$C_g = C_0 + \left\{ P\big(E_{componentsDamage}\big) * C_{consequenceOfDamagePriorToAdaptiveOptionsByActions_{\{a_1,.,a_i,.,a_{N_a}\}}} \right\} +$$

$$\left\{ \sum_{i=1}^{N_a} \left[ C_{setup_{action_{a_i}}} + \left\{ \left( \big(1 - P(E_{pa})\big) * C_{action_{a_i}} \right) + \left( P\big(E_{da_i}\big) * C_{consequence_{action_{a_i}}} \right) \right\} \right] \right\}$$

Where:
- $C_0$: initial cost of the whole infrastructures from the design stage until the initial service and use
- $N_a$: number of adaptive options, in order to solve the disturbance of the service (traffic, etc)
- $C_{setup\_a_i}$: Cost of the adaptive option i.e. design, staff, equipment, overheads, and daily service
- $C_{a_i}$: socio-economic consequences of non-efficiency of the adaptive option (overcome the disturbance, consider the public perception…)
- $C_{consequenceOf\_a_i}$: indirect or direct socio-economic impact of the adaptive option secondary effects

It is worth to notice that the modelling described above concerns:
- The effectiveness of action as $P\big(E_{pa}\big)$
- The effort of action as $C_{setup_{action_{a_i}}}$
- The damage of action as $P\big(\{\bar{E}_{da}|E_{pa}\}\big)$

Therefore, the part concerning the damage on internal infrastructures components is partly described through the loss of utility function. This damage as well as the transformation of the weights and metrics, presented Tables 15-18, will be
normalized and transformed into objective probabilities. This process is still under development and will be further detailed in an upcoming paper.



## 6 Discussion

### 6.1 Resilience Assessment

The assessment framework replied to the method presented in this study aims precisely at the indicators creation of a CI in a
defined scenario. This approach, based on a scenario, considering both consequences and optimisation actions, allows
studying a CI facing disasters with a global perspective. The objects of study, both disaster and infrastructure, remain
unchanged, and the values of resilience, criteria levels, and indicators change as suggested alternative roads are
implemented. Thus, the scenarios with different alternative roads could be assessed to find the better one. Furthermore,
under other optimisation actions, like "Construction of temporary bridges over flooded sections" or "Creating dams", the
sub-criteria and indicators relating to "action" should be modified. The problem then arises that the values of resilience or
general criteria, assessed by different indicators and sub-criteria, could not be compared. It results in the meaningless of the
values of resilience and general criteria in the indicators-based assessment suggested in this study. However, in practice,
their value, while important, is not the only significant part of the decision-making process, because resilience and general
criteria are too abstract and do not contain concrete information. Only with sub-criteria and indicators in place, managers
will be able to understand the content of each scenario in its entirety. Imagine now that two optimisation actions are an
option, "Creating dams" (A), and "Suggesting alternative roads" (B). Option A has a much higher resilience value than B,
since in the scenario where A is implemented, there is no significant "damage to internal components", and the
"effectiveness of action" is high even though the "effort for actions" and "damage of action are both high". Based on this
information, the choice of A is highly probable. Once further analysis of the sub-criteria and indicators reveals that the
concrete resource consumption of A is much higher than the city of Nantes can sustain, the choice would then be completely
reversed. Therefore, the analysis of the concept of resilience (definition, phenomena, aspects, etc.) and general criteria from
Yang et al. (2023) principally contribute to the design of sub-criteria and indicators, which play a key role in practice
management.

Multi-criteria and numerous indicators increase the complexity of practice to a certain extent. Nevertheless, there is no doubt
that the resilience of modern infrastructure is a complex object, but not a complicated one. A complicated object, i.e. one
with a certain amount of disorder, can be simplified, whereas a complex object should not be simplified. "Complexity varies
according to a number of parameters, including the multiple uses to which it is put, the number of participants involved, its
geographical dispersion, and the spatial and temporal scales considered" (Barroca et al., 2016). Consequently, a complex
indicators system accompanied by multi-criteria seems inevitable for CIs resilience assessment.





## 6.2 Limitations and Future Works

Many existing theories or models for CIs resilience assessment are valuable, although they differ in the disciplines and perspectives of this study. Nevertheless, the present study insists that, for resilience theory to become practical, it is
necessary to consider the cost-effectiveness and negative effects of the operation. Moreover, another key of resilience operationalisation is the uniqueness of each case that could be realised by specific sub-criteria and indicators. Just as teaching a man to fish, rather than simply giving him fish. Rather than predefining sub-criteria or indicators for all potential resilience scenarios of CIs resilience, the guide for indicators creation in this study provides enables users to design specific sub-criteria and indicators based on concrete situations. The methodology, therefore, provides a wide margin of autonomy
for managers and policymakers who have the responsibility for building CIs resilience and need support and guidance to operationalize the resilience-building process. Meanwhile, however, the autonomy of this guide can also be interpreted as a weakness. Managers' experience or knowledge may be so limited that they overlook invisible factors. From a holistic perspective, a collaborative multi-stakeholder exchange can reduce this shortcoming, whereas a significant investment of human resources at the same time may reduce the cost-benefit of collaborative management. Research in the field of
management is therefore needed for a better application of designed indicators systems.

Another limitation of this guide refers to the suggested method for data collection. As it is based on existing available resources, for instance in the presented example, few indicators relating to optimisation are applicable due to le lack of appreciable references or local data. Road infrastructures require the management of a large quantity of varied data
(topographical, geospatial, geometric, etc.), which is often available in heterogeneous formats. Intelligent digital systems can improve data collection and integration. However, the construction and maintenance of digital data of road infrastructure in Europe are not enough due to an insufficient level of cooperation, inadequate information management and limited investment in research, technology and development (UNECE, 2021). Without true historical data, professional and particular simulation models, for example by digital twin, would be acceptable. However, a specific model targeting given scenarios
that enables producing useful data resources for practice management has large time-consuming and high investment. It is instead less effectivity and cost-efficient. Potential challenges relate to effective and convenient ways of reference and data collection. On the other hand, for data managers, data resource building could take place from possible indicators. For serving the important indicators without available data, creating useful data resources presents a key task for local data institutions for the purpose of a continuous assessment.

## 7 Conclusion

Focusing on the resilience assessment of critical infrastructures, and in order to address a current challenge in resilience studies, the operationalisation of resilience assessment, this study develops a step-by-step guide of indicators creation in considering both positive and negative effects of optimisation actions that could be implemented. Three keys factor have



been identified for indicator creation: criteria setting, indicators setting with reference definition, and data collection. The
criteria setting in this study relies on the Multi-Criteria Framework (MCF) developed by Yang et al. (2023), which aims at
defining specific criteria depending on the real situation. Moreover, this MCF contributes to the operationalisation of
resilience assessment through its perspective of management sciences, particularly the criteria relating to optimisation
actions. Indicator setting refers to, in the absence of existing usable indicators, the manager could create particular indicators
based on the analysis of information dimensions (spatial, temporal, quantitative, and qualitative), and indicators reference
that is considered rulers for measuring criteria by indicators. Data collection respects three principles: relevance,
adaptability, and usability. CIs managers could benefit from the results of the indicators created through this guide could
during the decision-making process, as it is a multiple-criteria approach developed for allowing consideration of various
interests of stakeholders.

**Data availability**

All raw data can be provided by the corresponding authors upon request.

**Author contributions**

Conceptualization, Z.Y. and B.B.; methodology Z.Y., K.L.; investigation Z.Y.; writing—original draft preparation Z.Y. and
A.M.; writing—review and editing B.B., A.M. and K.L.; visualization, Z.Y; Data Curation, H.D and L.L; supervision B.B.
All authors have collaborated, read and agreed to the published version of the manuscript.

**Competing interests**

The authors declare that they have no conflict of interest.

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

**Appendix A**

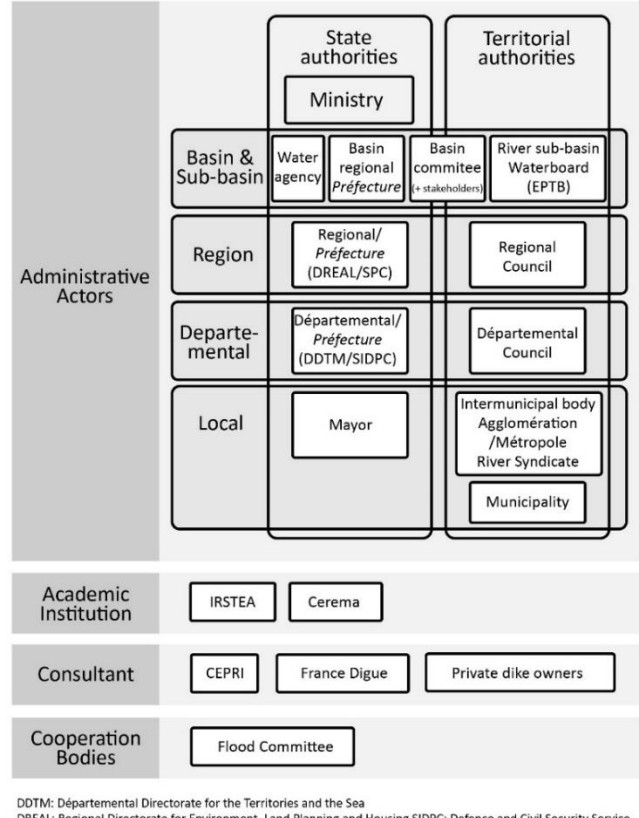

DDTM: Départemental Directorate for the Territories and the Sea
DREAL: Regional Directorate for Environment, Land Planning and Housing SIDPC: Defence and Civil Security Service
SPC: Regional Flood Forecasting Service
EPTB: River sub-basin Water Board
IRSTEA: Institut national de recherche en sciences et technologies pour l'environnement et l'agriculture
Cerema: Centre d'études et d'expertise sur les risques, l'environnement, la mobilité et l'aménagement
CEPRI: Centre Européen de prévention de Risque d'Inondation

**Fig. A1. Involved actors of flood management in France, source: Larrue et al. (2016) and Yang et al. (2021).**






**Fig A2. Involved actors of road infrastructure management in France, source: Yang et al. (2021).**

**Appendix B**

**Table B 1. Potentially usable open data websites, created by auteurs.**

| Organisations | Potentially applicable data | Link |
|---|---|---|
| Institut géographique national (IGN) | Geographic data in France | https://geoservices.ign.fr/catalogue |
| Data.gouv | Public data from the French State | https://www.data.gouv.fr/fr/ |
| Institut national de la statistique et des études économiques (INSEE) | Statistics and economic studies collect, produce, analyse and disseminate information on the French economy and society. | https://www.insee.fr/fr/accueil |
| Ville de Nantes, Nantes métropole | Open public data provided by the City of Nantes and Nantes Métropole. | https://data.nantesmetropole.fr/pages/home/ |
| CatNat | Database of natural disasters worldwide since 01/01/2001<br>Database of recognition/non-recognition of natural disasters by commune since 1982<br>Database of Natural Risk Prevention Plans (surveyed, prescribed or approved) by municipality<br>Database of local emergency plans (Plans Communaux de Sauvegarde) by municipality<br>Database of Municipal Information Dossiers on Major Risks<br>Flood Zones Atlas database by municipality<br>Flood Risk Territories database by municipality | https://www.catnat.net/nos-bases-de-donnees |
| Climate central | An interactive map showing areas threatened by sea level rise and coastal flooding. Combining the most advanced global model of coastal elevations with the latest projections for future flood levels. | https://coastal.climatecentral.org/ |
| Géorisque | Database on all types of risk in France | https://www.georisques.gouv.fr/donnees/bases-de-donnees |
