# Peer review of "Critical Infrastructures Resilience: A Guide for Building Indicator Systems"

_EGUsphere, 2024_

## Author Comment (AC1)

| N° | Comment | Revision |
|---|---|---|
| 1 | It is recommended that the author revise the title. In the current title, the two contents do not seem to be closely related, causing trouble for readers to obtain the information of this manuscript in the first time. | **Modified (red)**

Critical Infrastructures Resilience: A guide for Building Indicators Systems
Based on a Multi-Criteria Framework with a Focus on Optimising Actions |
| 2 | The structure of the manuscript is extremely unreasonable. Some sections have the same title, the content is lengthy, the focus is not highlighted, and the structure is not conducive to reading. It is recommended that the author make a complete restructuring of the manuscript. | **Added and modified (red)**

**2 Research Methodology and Structure**

In pursuit of the study's objective, the current inquiry arises: how to develop a framework that enables CIs stakeholders to build a specific indicators system for assessing the resilience of studied CIs. Practical guides should include guidance on practical steps, resources, and tools. Therefore, the steps, as well as the information and advice for building indicators systems, are anticipated to be developed in the objective guide. One fundamental query necessitates deliberation: what achieves should the steps assist the user in accomplishing? For building an indicators system, the identification of criteria, indicators, and data is considered basic, as they are the indispensable contents of an indicators system. Many studies, such as those carried out by Van Bueren and Blom (1997), Prabhu et al. (1999), and Mendoza et al. (2000), consider that the usable criteria and indicators adapted to the specific needs of stakeholders are the key to applying indicator systems to practical management. Moreover, several studies believe that data analysis should not be missed during the indicators-based assessment (Vogel, 1997; 1996; Prabhu et al. 1999; Cutter, 2016; CORDIS-Smart Resilience Indicators for Smart Critical Infrastructures, 2018; Balaei et al., 2018). Therefore, the first part of this research should be to provide an interpretation of the three basic key factors in conjunction with relevant research materials: criteria, indicators, and data. The steps needed to set these factors could be therefore identified (Fig.2).

**Fig. 2. Methodology and structure of the present study.**

The second research part concerns designing the needed steps, which are identified in the first part as vital. Thus, this part will discuss some existing frameworks or theories about Moreover, for these steps to be better applied in practice, the steps designed in this guide should be clearly described and preferably accompanied by schematic diagrams. The designing steps are combined in the objective guide, which should provide detailed assistance to users in C&I setting and data collection.

After designing steps, this study applies them to a French critical infrastructure to build an indicators system that can assess resilience during urban flooding. The example relies on the Nantes Ring Road (NRR) system with the participation of a local management |

organisation-Direction interdépartementale des routes Ouest (DIRO) in charge of the road networks of Nantes City in France. This application example involves 62 676 data for traffic flow from DIRO and more than 15 000 data of road infrastructures from BDTOPO of National Geographic Institute (IGN).

Therefore, this study is divided into several sections for implementing the parts (Fig. 2). Section 3 will discuss the three indispensable key factors for building an indicators system: Criteria, Indicator, and Dada. Section 4 designs a step-by-step guide that helps users build an indicators system based on their particular situations. Section 5 will illustrate how to use this developed guide to build an indicators system through an example. Section 6 discusses the practical use, and the limitation of this guide, and shows a comprehensive assessment process (including resilience and indicator assessment phases in Fig.1) in using the built indicators systems in Section 5.

**3 Part 1: Keys Factors Presentation**

**3.1 Criteria**

**3.2 Indicators**

**3.3 Data**

**3.4 Result of part 1: needed steps**

**4. Part 2: Steps Designing**

**4.1 Specific criteria setting**

**4.2 Possible indicator setting and reference definition**

**4.3 Indicators selection through the availability of data**

**4.4 Result of part 2: Step-by-step guide**

**5 Example of Guide Usage**

**5.1 Criteria setting**

**5.2 Possible Indicators setting**

**5.3 Available data analysis**

**5.4 Result of part 3: Indicators system for studied CI**

**Added (red)**

**6. Discussion**

**6.1 A practical and operational guide**

Multi-criteria and numerous indicators increase the complexity of practice to a certain extent. Nevertheless, there is no doubt that the resilience of modern infrastructure is a complex object, but not a complicated one. A complicated object, i.e. one with a certain amount of disorder, can be simplified, whereas a complex object should not be simplified. "Complexity varies according to a number of parameters, including the multiple uses to which it is put, the number of participants involved, its geographical dispersion, and the spatial and temporal scales considered" (Barroca et al., 2016). Consequently, a complex indicators system accompanied by multi-criteria seems inevitable for CIs resilience assessment. The more complex an indicators system, the more it requires detailed knowledge of local condition. At the same time, the higher the need to increase the autonomy of local managers, which the developed guide in this study provides.

Many existing theories or models for CIs resilience assessment are valuable, although they differ in the disciplines and perspectives of this study. Nevertheless, the present study insists that, for resilience theory to become operational, it is necessary to consider the cost-effectiveness and negative effects of the operation. Moreover, another key of resilience operationalisation is the uniqueness of each case that could be realised by specific sub-criteria and indicators. Just as teaching a man to fish, rather than simply giving him fish. Rather than predefining sub-criteria or indicators for all potential resilience scenarios of CIs resilience, the guide for indicators creation in this study provides enables users to design specific sub-criteria and indicators based on concrete situations. The design guide, therefore, provides a wide margin of autonomy for managers and policymakers who have the responsibility for building CIs resilience and need support and guidance to operationalise the resilience-building process. The autonomy also brings the possibility of continuous updating or optimising of the indicator system. Changes in the external environment may lead to changes in the setting and weighting of criteria, indicators. For example, the sub-criteria of "Environmental damage" and the indicator of "Additional $CO_2$ emission" have become important in recent years because of the development of environmental concern.

Meanwhile, the autonomy of this guide can also be interpreted as a weakness. Managers' experience or knowledge may be so limited that they overlook invisible factors. From a holistic perspective, a collaborative exchange between different stakeholders can reduce this shortcoming. The examples in this study demonstrate exactly the kind of co-operation between local operators, university scientists and local researchers. Whereas a significant investment of human resources at the same time may reduce the cost-benefit of collaborative management. Research in the field of management is therefore needed for a better application of designed indicators systems.

In addition, the designed guide promotes the practical use of resilience indicators and further contributes to the operationalisation of CIs resilience assessment. Operationalising the concept of "resilience" is considered a major milestone that contributes to the risks management for CIs, even for cities, and the interactions required to build and sustain it. The current studies of the CIs resilience aim to develop more effective and sustainable infrastructure management strategies for CIs through the concept of "resilience". In other words, one of the desired developments in resilience research is to put resilience-based theories, tools, and models into practice and make them useful and operational in risks management. However, despite existing efforts, the obstacle to operationalising the CIs resilience concerns two major limitations: 1) the absence of applied tools; 2) the lack of an organisational aspect (Weichselgartner and Kelman, 2015; Hernantes et al., 2019 ;Heinzlef et al., 2022; Rød et al., 2020; Yang et al., 2023a). The guide designed in this study is firstly a tool that can be applied for concrete scenarios, as demonstrated by the case studies presented. The fact that the guide helps setting criteria based on operational perspectives is also emphasised several times. Operationalisation through this developed guide consists of giving CIs resilience a practical and operational meaning, transforming it into an object of practical value, in the broader sense of 'use'.

**6.2 Assessment demonstration**

**6.2.1 Criteria & Indicators weighting**

**6.2.2 Assessment methods and results**

**6.3 Limitation**

---

## Author Comment (AC2)

| | 1. **Paper's structure** |
| | 3. **Example demonstration** |
|---|---|
| **Comment** | One major comment is that this paper could greatly benefit from a holistic illustrative example to apply the framework the author proposed. This example could serve as a powerful tool to demonstrate the practical application of the framework, potentially inspiring the authors to further develop their work.

Figure 2, why does it need a guide design?  Why can we not develop criteria from key factors directly? Again, the audience will be more convinced if the paper has an example. |
| **Reply** | The structure of this paper is unclear due to the step of "step designing". We suggest combining sections for "key factors identification" and "step designing". Thus, the designed steps, for the "Criteria & Indicators" setting and data analysis, could be presented directly with the identification of key factors (Criteria, indicators, and data). In the new version, the result of section 3 is therefore the development of a guide for building indicator systems. Section 4 shows then a big example for demonstrating the use of the developed guide.
Some example demonstrations are not necessary. Therefore, in the reviewed version, there is only a big example that includes all the steps shown in the developed guide. |
| **Revision 1**

**Section 2: Research Methodology and Structure** | To achieve the objectives of this study, … Therefore, criteria, indicators and data are the indispensable contents of an indicators system. For building an indicator system, the setting of Criteria & Indicators (C&I), and the collection of data are considered basic.  This research could start with a presentation of the three basic key factors (criterion, indicator and data). Then, the main research work is designing the steps for C&I setting and data collection (Fig.2). Moreover, for these steps to be better operational in practice, the steps designed in this guide should be clearly described and preferably with the support of schematic diagrams.

[Figure]

**Fig. 2. Methodology and structure of the present study.**

In the second part, this study applies the designed steps to a French critical infrastructure to build an indicator system that can assess resilience during urban flooding (Fig.2). The example focuses on the Nantes Ring Road (NRR) network, the investigation of which was assisted by a local management organisation, Direction interdépartementale des routes Ouest (DIRO) that is in charge of the road networks of Nantes City in France. This example involves 62 676  traffic flow data from DIRO, and over 15 000 road infrastructure data from French National Geographic Institute (IGN).

The present paper is divided into several sections. Section 3 will (Fig.2) develop a step-by-step guide that enables CIs managers building indicator systems for their particular studied cases. Section 4 (Fig.2) will illustrate how to use this developed guide to build an indicators system through an example focusing on Nantes Ring Road. Section 5 discusses the contributions, and limitations of this guide, and shows an assessment process (including resilience and indicator assessment phases in Fig.1) in using the built indicator system in Section 4. |

| **Revision 2**

**paper's structure** | 1 Introduction
2 Research Methodology and Structure
3 Part 1: Guide's Steps Designing
    3.1 Specific criteria setting
        3.1.1 Direct and indirect damages
        3.1.2 Effectiveness and efforts of actions
    3.2 Indicators setting and references definition
    3.3 Verification of available data
    3.4 Result of part 1: Step-by-step guide
4 Example of Guide Usage
    4.1 Criteria setting
      Initial scenario
      Continuous scenarios
    4.2 Possible Indicators setting
    4.3 Available data analysis
    4.4. an indicator system for studied example case
5. Discussion
    5.1 A practical and operational guide
    5.2 Assessment demonstration
      5.2.1 Criteria & Indicators weighting
      5.2.2 Assessment methods and results
    5.3 Limitation
    6 Conclusion |

| **2. Synonyms** | |
|---|---|
| **Comment** | Another big issue is that the novelty of this paper is not clear. The word "operationalizing" may not be the most appropriate term. The authors may want to consider using "application" or "implementation". However, without a clear and compelling illustrative example, it becomes challenging to substantiate the novelty of this paper as the authors proposed. This underscores the importance of revising and improving the argumentation to ensure clarity. |
| **Reply** | We agree with you about the confused use of "operationalisation", "application" and "implementation". This paper wants to discuss two topics: the application of indicators-based assessment for critical infrastructure resilience; and the implementable actions identified through the Behind the Barriers model. However, the initial paper did not well distinguish these terms. This problem has been resolved in the new version. Since the focus of the paper is on indicator systems built by a developed guide, one discussion refers to the contribution of developed guide and indicator systems to the application of CIs resilience assessment. |
| **Revision 1**

**Abstract** | Criteria and indicators are frequently used for assessing the resilience of Critical Infrastructures (CIs). Moreover, to generate precise information on conditions, the assessment designed for CIs resilience could rely on indicator systems. However, few practical tools exist for guiding CIs managers to build specific indicator systems in considering local realities. Therefore, the main objective of this study is to develop a step-by-step guide that contains guidance on operational steps and required resources for Criteria & Indicators setting, references definition, and data collection. This guide enables CIs managers to build systems of indicators adapted to different realities. This guide could assist CIs managers in their decision-making process, as it is structured based on a multi-criteria framework that takes into account the cost-benefits and side effects of implementable actions. This guide could furthermore advance the application of indicator-based CIs resilience assessment in practical management. In addition, this study provides an example to |

| | |
|---|---|
| | demonstrate how to use this guide. This example is based on a given scenario for the Nantes Ring Road (NRR) network: when the ring road is flooded and closed, the road network manager suggests alternative roads to the public. An indicator system, consisting of 4 criteria, 7 sub-criteria and 11 indicators, could be built for this scenario through the developed guide. This example relates to criteria and indicators in technical, social, and environmental dimensions, and involves 62 676 data. |
| **Revision 2**

**Last paragraph of section 1: Introduction** | Indicator systems building involves criteria setting. Criteria serve as characters or signs making a judgment of appreciation. From an operational perspective, multi-criteria analysis allows CIs managers to keep holistic thinking that balances the various advantages and disadvantages (Yang et al., 2023, b). However, many studies about CIs resilience criteria setting have focused on abstract capabilities related to resilience, but have overlooked the fact: the benefits, costs or impacts of implementable actions for every CIs manager are critical. The lack of discussion about the effects of implementable actions causes the application difficulties of CIs resilience assessment in practical management. Therefore, the developed guide for building indicator systems should consider a criteria-setting framework involving implementable actions. The ways for multi-criteria setting involving implementable actions should be added in the objective guide of this present study. |
| **Revision 3**

**The second and third paragraphs of section 3.1: Specific criteria setting** | Assessments consisting of Criteria & Indicators (C&I) … 
[revised manuscript text omitted]

| 4. Figures' illustration and description | |
|---|---|
| **Comment** | Some figures are not clear. For example, in Figure 1, it would be better if the lines had an arrow pointing from right to left. Some figures need more associated explanations. For example, in Figure 3: Why does it need the layer of aspects? |
| **Reply** | Some example demonstrations are not necessary. Therefore, in the reviewed version, there is only a big example that includes all the steps shown in the framework. |
| **Revision 1**

**Section 1 introduction**
**Fig. 1** |
[Figure]

[revised manuscript text omitted]

---

## Author Comment (AC3)

| | |
|---|---|
| **Comment** | A suggestion is to clarify how the author's work benefits from and differs from existing related studies in the Introduction Section. |
| **Reply** | The presented study aims to meet one current requirement for the indicators-based resilience assessment in practical management:  a guide for indicator systems building. Despite the existence of a large number of resilience assessment indicators, very little research develops guidance to help managers create specific indicators themselves. However, indeed, we did not point out how this presented study differs from other available research. Therefore, we have added this part as you suggested, and please find the detailed revisions in the attached document (revision 3). All your comments are in green and all highlight revisions are in red. |
| **Revision**

**Section 1: introduction** | A single indicator can rarely provide useful information. To generate increasingly precise information on conditions, the assessment designed for a complex system could rely on indicator systems (Shavelson et al., 1990). An indicator system should contain numerous specific indicators that are associated with concrete conditions, requirements, or situations. These specific indicators could not be set without consideration of the realities of each particular studied case. Thus, it necessitates practical tools that enable CIs managers to set their specific indicator system for their particular studied case, without providing directly pre-defined indicators. As argued by Shavelson et al. (1991) "no indicator system could accommodate all of the potential indicators identified by a comprehensive process and remain manageable". A desirable hazard-related indicators tool should be simple and flexible, adapting itself to different case studies and different kinds of users (Barroca et al. 2016). Even though existing CIs resilience assessments by indicators are diverse and multidisciplinary, few tools exist for guiding CIs managers build specific indicator systems in regarding realities. For example, Yang et al. (2023, a) review 68 scientific papers relating to indictors-based assessments for CIs resilience. Several papers reviewed by Yang et al. (2023, a) present assessments based on a large number of systemic indicators: Fisher et al. (2010), Hromada and Lukas (2012), Petit et al. (2013), Bialas (2016), Upadhyaya et al. (2018), De Vivo et al. (2022). However, all these papers directly show the suggested indicators without describing the detailed steps to set them. Therefore, the main objective of this study is to provide a guide for CIs managers to enable them to build specific indicator systems for their particular studied cases. |

---

## Author Comment (AC4)

| Paper's structure | |
|---|---|
| **Comment** | why does it need a guide design? Why can we not develop criteria from key factors directly? Again, the audience will be more convinced if the paper has an example.

One possible solution is that the authors do not have to propose a small example for each section (such as lines 132-145) but use a big example to include all the steps shown in the framework, and then echo back which part is associated to which step. This could help avoid the body text looking tedious and repetitive, and instead make it clear and concise. |
| **Reply** | The structure of this paper is unclear due to the step of "step designing". We suggest combining sections for "key factors identification" and "step designing". Thus, the designed steps, for the "Criteria & Indicators" setting and data analysis, could be presented directly with the identification of key factors (Criteria, indicators, and data). In the new version, the result of section 3 is therefore the development of a guide for building indicator systems. Section 4 shows then a big example for demonstrating the use of the developed guide. |
| **Revision 1**

**Section 2: Research Methodology and Structure** | **2 Research Methodology and Structure**

To achieve the objectives of this study, … Therefore, criteria, indicators and data are the indispensable contents of an indicators system. For building an indicator system, the setting of Criteria & Indicators (C&I), and the collection of data are considered basic. This research could start with a presentation of the three basic key factors (criterion, indicator and data). Then, the main research work is designing the steps for C&I setting and data collection (Fig.2). Moreover, for these steps to be better operational in practice, the steps designed in this guide should be clearly described and preferably with the support of schematic diagrams.

[Figure]

**Fig. 2. Methodology and structure of the present study.**

In the second part, this study applies the designed steps to a French critical infrastructure to build an indicator system that can assess resilience during urban flooding (Fig.2). The example focuses on the Nantes Ring Road (NRR) network, the investigation of which was assisted by a local management organisation, Direction interdépartementale des routes Ouest (DIRO) that is in charge of the road networks of Nantes City in France. This example involves 62 676 traffic flow data from DIRO, and over 15 000 road infrastructure data from French National Geographic Institute (IGN).

The present paper is divided into several sections. Section 3 will (Fig.2) develop a step-by-step guide that enables CIs managers building indicator systems for their particular studied cases. Section 4 (Fig.2) will illustrate how to use this developed guide to build an indicators system through an example focusing on Nantes Ring Road. Section 5 discusses the contributions, and limitations of this guide, and shows an assessment process (including resilience and indicator assessment phases in Fig.1) in using the built indicator system in Section 4. |

| | |
|---|---|
| **Revision 2**

**paper's structure** | 1 Introduction
2 Research Methodology and Structure
3 Part 1: Guide's Steps Designing
      3.1 Specific criteria setting
          3.1.1 Direct and indirect damages
          3.1.2 Effectiveness and efforts of actions
      3.2 Indicators setting and references definition
      3.3 Verification of available data
      3.4 Result of part 1: Step-by-step guide
4 Part 2: Example of Guide Usage
      4.1 Criteria setting
       Initial scenario
       Continuous scenarios
      4.2 Possible Indicators setting
      4.3 Available data analysis
      4.4. an indicator system for studied example case
5. Discussion
      5.1 A practical and operational guide
      5.2 Assessment demonstration
          5.2.1 Criteria & Indicators weighting
          5.2.2 Assessment methods and results
      5.3 Limitation
    6  Conclusion |

| **Use of some similar terms** | |
|---|---|
| **Comment** | The word "operationalizing" may not be the most appropriate term. The authors may want to consider using "application" or "implementation". However, without a clear and compelling illustrative example, it becomes challenging to substantiate the novelty of this paper as the authors proposed. This underscores the importance of revising and improving the argumentation to ensure clarity |
| **Reply** | The study is confusing in its use of 'operationalisation', 'application' and 'implementation'. This paper wants to discuss two topics: the application of indicators-based assessment for critical infrastructure resilience; and the implementable actions identified through the Behind the Barriers model. However, the initial paper did not well distinguish these terms. This problem has been resolved in the new version. Since the focus of the paper is on indicator systems built by a developed guide, one discussion refers to the contribution of developed guide and indicator systems to the application of CIs resilience assessment. |
| **Revision 1**

[revised manuscript text omitted]

---

## Author Comment (AC5)

| | |
|---|---|
| **The focus of this manuscript.** | |
| **Comment** | It is recommended that the author revise the title. In the current title, the two contents do not seem to be closely related, causing trouble for readers to obtain the information of this manuscript in the first time. |
| **Reply** | The focus and the title of this study should be redefined. Thus, the topic of the study has been corrected as follows: Critical Infrastructures Resilience: A Guide for Building Indicator Systems. The focus of this study is to develop a guide, which enables Critical Infrastructures stakeholders to build their specific indicator systems. This guide should provide practical steps on criteria setting, indicator setting, and data collection. The relevant detailed revision can be found in the attached document. |
| **Revision**

**Section 1:**
**Introduction** | **1 Introduction**

The research for Critical Infrastructures (CIs) goes across disciplines, sectors, and scales, as the disruption or destruction of CIs would have a significant cross-border impact on human society. However, CIs are vulnerable to natural and technological hazards worldwide. The concept of "Resilience", presented as an inherent attribute of a system addressing external hazards, has developed rapidly recently in the field of  CIs management. Meanwhile, resilience assessments have become an important aspect for CIs management. An efficient resilience assessment could integrate a set of key concepts and provide alternative ways of thinking about and practicing resource management (Resilience Alliance). Moreover, the assessment of CIs resilience is frequently based on indicators (Hosseini et al., 2016; Mebarki, 2017; Cantelmi et al., 2021). Indicator-based resilience assessment could be simply summarised as a process consisting of three factors and two phases, as shown in Fig. 1 (Yang et al., 2023, a):

- Resilience assessment: a process in which resilience values are obtained by usable indicators;
- Indicator assessment: a process in which indicator values are obtained by reliable data.

The principle of indicator-based assessment is transforming from data, to indicators, and from indicators to knowledge or goal. Available methods for both resilience and indicators assessments are diverse and multidisciplinary and could be quantitative, qualitative, and semi-quantitative (Hosseini et al., 2016; Mebarki, 2017; Yang et al., 2023, a).

**Fig. 1. Indicator-based Resilience Assessment, source: Yang et al. (2023, a).**

A single indicator can rarely provide useful information. To generate increasingly precise information on conditions, the assessment designed for a complex system could rely on indicator systems (Shavelson et al., 1990). An indicator system should contain numerous specific indicators that are associated with concrete conditions, requirements, or situations. These specific indicators could not be set without consideration of the realities of each particular studied case. Thus, it necessitates practical tools that enable CIs managers to set their specific indicator system for their particular studied case, without providing directly pre-defined indicators. As argued by Shavelson et al. (1991) "no indicator system could accommodate all of the potential indicators identified by a comprehensive process and remain manageable". A desirable hazard-related indicators tool should be simple and flexible, adapting itself to different case studies and different kinds of users (Barroca et al. 2006). Even though existing CIs resilience assessments by indicators are diverse and multidisciplinary, few tools exist for guiding CIs managers build specific indicator systems in |

regarding realities. For example, Yang et al. (2023, a) review 68 scientific papers relating to indictors-based assessments for CIs resilience. Several papers reviewed by Yang et al. (2023, a) present assessments based on a large number of systemic indicators: Fisher et al. (2010), Hromada and Lukas (2012), Petit et al. (2013), Bialas (2016), Upadhyaya et al. (2018), De Vivo et al. (2022). However, all these papers directly show the suggested indicators without describing the detailed steps to set them. Therefore, the main objective of this study is to provide a guide for CIs managers to enable them to build specific indicator systems for their particular studied cases.

Indicator systems building involves criteria setting. Criteria serve as characters or signs making a judgment of appreciation. From an operational perspective, multi-criteria analysis allows CIs managers to keep holistic thinking that balances the various advantages and disadvantages (Yang et al., 2023, b). However, many studies about CIs resilience criteria setting have focused on abstract capabilities related to resilience, but have overlooked the fact: the benefits, costs or impacts of implementable actions for every CIs manager are critical. The lack of discussion about the effects of implementable actions causes the application difficulties of CIs resilience assessment in practical management. Therefore, the developed guide for building indicator systems should consider a criteria-setting framework involving implementable actions. The ways for multi-criteria setting involving implementable actions should be added in the objective guide of this present study.

| Discussion | |
| --- | --- |
| **Comment** | The discussion part is weak, does not grasp the focus of this manuscript's work, and does not incorporate enough previous work. |
| **Reply** | We have optimised the discussion section regarding the use of the developed guide and its limitations, to better relate the discussion to the objectives of the study. Additionally, the demonstration of resilience assessment has been moved to the discussion section and appendixes, as it is not the primary focus of this study. The Emphasis of this study has been placed on designing practical steps for criteria setting, indicator setting, and data collection. The relevant detailed revision can be found in the attached document. |
| **Revision**

[revised manuscript text omitted]

---

## Author Response (AR1)

| | |
|---|---|
| **Issue** | **The focus of this manuscript: build indicator systems for resilience assessments**
**Research gap and research questions**
**Discussion focusing on the application of resilience assessment**
**Confused use of "operationalisation", "application" and "implementation"** |
| **Comment** | **Anonymous Referee #1:**
It is recommended that the author revise the title. In the current title, the two contents do not seem to be closely related, causing trouble for readers to obtain the information of this manuscript in the first time.

The discussion part is weak, does not grasp the focus of this manuscript's work, and does not incorporate enough previous work. The current Discussion section could not provide valuable information to readers.

**Anonymous Referee #2:**
Another big issue is that the novelty of this paper is not clear. The word "operationalizing" may not be the most appropriate term. The authors may want to consider using "application" or "implementation". However, without a clear and compelling illustrative example, it becomes challenging to substantiate the novelty of this paper as the authors proposed. This underscores the importance of revising and improving the argumentation to ensure clarity.

**Yuxuan Yao:**
A suggestion is to clarify how the author's work benefits from and differs from existing related studies in the Introduction Section.

**Editors:**
the comments regarding previous research and novelty of the study must be addressed by highlighting the research gap and research questions. The reference list is good, but please pay attention also to formatting |
| **Reply** | The focus and the title of this study has been redefined based on the identified research gap.

Despite the existence of a large number of resilience assessment indicators, very little research develops guidance to help managers create specific indicators themselves. Thus, The presented study aims to meet one current requirement for the indicators-based resilience assessment in practical management: a guide for indicator systems building. The topic of the study has been corrected as follows: Critical Infrastructures Resilience: A Guide for Building Indicator Systems. The focus of this study is to develop a guide, which enables Critical Infrastructures stakeholders to build their specific indicator systems. This guide should provide practical steps on criteria setting, indicator setting, and data collection. Indeed, we did not point out research gap in the original version. Therefore, we have added this part in the "introduction".

In addition, we agree that it exists a confused use of "operationalisation", "application" and "implementation". This paper wants to discuss two topics: the application of indicators-based assessment for critical infrastructure resilience; and the implementable actions identified through the Behind the Barriers model. However, the initial paper did not well distinguish these terms. This problem has been resolved in the new version. Since the focus of the paper is on indicator systems built by a developed guide, one discussion refers to the contribution of developed guide and indicator systems to the application of CIs resilience assessment. |

[revised manuscript text omitted]

| Issue | Paper's structure
Example demonstration |
|---|---|
| Comment | **Anonymous Referee #1:**
The structure of the manuscript is extremely unreasonable. Some sections have the same title, the content is lengthy, the focus is not highlighted, and the structure is not conducive to reading. It is recommended that the author make a complete restructuring of the manuscript.

**Anonymous Referee #2:**
One major comment is that this paper could greatly benefit from a holistic illustrative example to apply the framework the author proposed. This example could serve as a powerful tool to demonstrate the practical application of the framework, potentially inspiring the authors to further develop their work.

Figure 2, why does it need a guide design?  Why can we not develop criteria from key factors directly? Again, the audience will be more convinced if the paper has an example. |
| Reply | The structure of this paper is unclear due to the step of "step designing". We suggest combining sections for "key factors identification" and "step designing". Thus, the designed steps, for the "Criteria & Indicators" setting and data analysis, could be presented directly with the identification of key factors (Criteria, indicators, and data). In the new version, the result of section 3 is therefore the development of a guide for building indicator systems. Section 4 shows then a big example for demonstrating the use of the developed guide.
Some example demonstrations are not necessary. Therefore, in the reviewed version, there is only a big example that includes all the steps shown in the developed guide.
In addition, since the assessment process is not the focus of this study, it is placed in the conclusions section. Details of the indicator assessment haves been places in Appendix C and Appendix D to reduce the number of pages. |
| Revision 1

Section 2: Research Methodology and Structure | To achieve the objectives of this study, … Therefore, criteria, indicators and data are the indispensable contents of an indicators system. For building an indicator system, the setting of Criteria & Indicators (C&I), and the collection of data are considered basic.  This research could start with a presentation of the three basic key factors (criterion, indicator and data). Then, the main research work is designing the steps for C&I setting and data collection (Fig.2). Moreover, for these steps to be better operational in practice, the steps designed in this guide should be clearly described and preferably with the support of schematic diagrams.

**Fig. 2. Methodology and structure of the present study, created by authors.**

In the second part, this study applies the designed steps to a French critical infrastructure to build an indicator system that can assess resilience during urban flooding (Fig.2). The example focuses on the Nantes Ring Road (NRR) network, the investigation of which was assisted by a local management organisation, Direction interdépartementale des routes Ouest (DIRO) that is in charge of the road |

| | |
|---|---|
| | networks of Nantes City in France. This example involves 62,676 traffic flow data from DIRO, and over 15,000 road infrastructure data from French National Geographic Institute (IGN).

The present paper is divided into several sections. Section 3 will (Fig.2) develop a step-by-step guide that enables CIs managers building indicator systems for their particular studied cases. Section 4 (Fig.2) will illustrate how to use this developed guide to build an indicators system through an example focusing on Nantes Ring Road. Section 5 discusses the contributions, and limitations of this guide, and shows an assessment process (including resilience and indicator assessment phases in Fig.1) in using the built indicator system in Section 4. |
| **Revision 2**

**paper's structure** | 1 Introduction
2 Research Methodology and Structure
3 Part 1: Guide's Steps Designing
    3.1 Specific criteria setting
        3.1.1 Direct and indirect damages
        3.1.2 Effectiveness and efforts of actions
    3.2 Indicators setting and references definition
    3.3 Verification of available data
    3.4 Result of part 1: Step-by-step guide
4 Example of Guide Usage
    4.1 Criteria setting
      Initial scenario
      Continuous scenarios
    4.2 Possible Indicators setting
    4.3 Available data analysis
    4.4. Result of part 2: an indicator system for studied example case
5. Discussion
    5.1 A practical and operational guide
    5.2 Assessment demonstration
        5.2.1 Criteria & Indicators weighting
        5.2.2 Assessment methods and results
    5.3 Limitation
  6 Conclusion |

| Issue | Figures' illustration and description |
|---|---|
| **Comment** | **Anonymous Referee #2:**
Some figures are not clear. For example, in Figure 1, it would be better if the lines had an arrow pointing from right to left. Some figures need more associated explanations. For example, in Figure 3: Why does it need the layer of aspects? |
| **Reply** | Some example demonstrations are not necessary. Therefore, in the reviewed version, there is only a big example that includes all the steps shown in the framework. |
| **Revision 1**

**Section 1 introduction Fig. 1** |
[Figure]

**Fig. 1. Indicator-based Resilience Assessment, source: Yang et al. (2023, a).** |
| **Revision 2**

**Section 2 Research Methodology and Structure**

**Fig. 3** | Assessments consisting of Criteria & Indicators (C&I) could provide a commonly agreed framework for articulating and defining expectations. There is a hierarchical structure for C&I based assessments (Fig. 3), firstly developed for forest sustainability assessment (Prabhu et al.,1996; Lammerts Van Bueren and Blom, 1997; Mendoza and Prabhu, 2010), today is also used in other disciplines (Montaño et al., 2006; Van Cauwenbergh et al., 2007; Koschke et al., 2012; Feiz and Ammenberg, 2017). This hierarchical structure is a common framework, in which a higher-level "goal" is divided into aspects or themes, which are in turn divided into criteria each with several indicators (Maggino, 2017). The assessment process (Fig3. Indicators-based assessment process) is from "indicators" to "goals", but criteria and indicators (Fig3. Criteria & Inidcators setting process) are set in the opposite direction. This means that the criteria and indicators are set based on certain important aspects of the assessed goal. Important aspects, in turn, are identified in terms of the definition and phenomenon of the assessed goal (Eurostat, 2014; Maggino, 2017). The aspects of the assessed goal may not be necessary for the assessment process, but they are important for criteria setting. In practical management, the criteria vary between different contexts. The designed criteria-setting steps in the present paper should enable managers to set specific criteria for adapting to different real cases.

**Fig. 3. A hierarchical structure in multi-criteria approaches for C&I-based assessment, adjusted from Yang et al. (2023, b).** |
| **Revision 3**

**Section 3.1.1 Direct** | The determination of significant damages is related to two criteria: "damage to internal components" and "damage of actions". Significant damages could be determined based on Form 1 introduced by Yang et al. (2023, b) (Fig. 5). This Form 1 can be considered as a process of setting specific sub-criteria under these two damage related criteria. According to Form 1, once the target CI (Fig.5. Affected system) has been defined, its four categories of components should be identified: function (seen as a type of component), collective human components, individual |

[revised manuscript text omitted]

---

## Author Response (AR2)

| To : Editor |
| --- |
| Dr. Maria Bostenaru Dan |

Subject: third revision of a manuscript entitled "Critical Infrastructures Resilience: A Guide for Building Indicator Systems. Based on a Multi-Criteria Framework with a Focus on Implementable Actions".

| First manuscript submitted on January 24 2024 | Second manuscript submitted on July 22 2024 |
| --- | --- |

Paris, August 2 2024

Dear Dr. Maria Bostenaru Dan,

Thanks a lot for your reply and for giving us suggestions to improve this manuscript.

Based on your comments, the third revision focuses mainly on the introduction. Firstly, it has been restructured and it now contains three paragraphs

1.      A short presentation of resilience assessment based on indicators.
2.      Research gap definition through a brief literature overview.
3.      Research questions, objectives, and assumptions.

Secondly, Figure 1 (original version) explaining the assessment process has been removed as its contents have been shown in Figure 3 and Figure 12 (revised version). This study concentrates on building indicator systems for resilience assessment. Therefore the processes of resilience assessment and indicators assessment have been moved to the discussion, i.e. section "5.2 Assessment demonstration".

In addition, some of the figures have been optimised, and grammatical mistakes have been corrected. Please find the detailed revisions in the following tables. All your comments are in green and all highlight revisions are in red.

Thanks again for your valuable comments provided.
Looking forward to receiving your reply.

Best regards,
Authors.

| Issue : Figure 1 | |
|---|---|
| **Comment** | While the figure in the introduction is enlighting, no figures shall be placed in the introduction. Their detailing, be it of the literature review or that related to indicators leading to this figure, shall be separate paragraphs. |
| **Reply** | Figure 1 (original version) explaining the assessment process has been removed as its contents have been shown in Figure 3 and Figure 12 (revised version). This study concentrates on building indicator systems for resilience assessment. Therefore the processes of resilience assessment and indicators assessment have been moved to the discussion, i.e. section "5.2 Assessment demonstration". |
| **Revision 1**

**Section 1: Introduction** | **Removed** *(red)*

[revised manuscript text omitted]